# Epigenetic reprogramming shapes the cellular landscape of schwannoma

S. John Liu [1,2,3,4], Tim Casey-Clyde [1,2,3,13], Nam Woo Cho[1,5,13],
Jason Swinderman[4,6], Melike Pekmezci [3], Mark C. Dougherty [7],
Kyla Foster [1,2,3], William C. Chen[1,2,3], Javier E. Villanueva-Meyer[8],
Danielle L. Swaney [9], Harish N. Vasudevan[1,2], Abrar Choudhury [1,2,3],
Joanna Pak[1,2,3,4], Jonathan D. Breshears[2], Ursula E. Lang[3,10], Charlotte D. Eaton[1,2,3],
Kamir J. Hiam-Galvez[5], Erica Stevenson[9], Kuei-Ho Chen[9], Brian V. Lien[2],
David Wu[2], Steve E. Braunstein[1], Penny K. Sneed[1], Stephen T. Magill[11], Daniel Lim[2],
Michael W. McDermott[12], Mitchel S. Berger[2], Arie Perry [2,3], Nevan J. Krogan [9],
Marlan R. Hansen[7], Matthew H. Spitzer [5], Luke Gilbert [4,6],
Philip V. Theodosopoulos[2] & David R. Raleigh [1,2,3] ✉

Mechanisms specifying cancer cell states and response to therapy are incompletely understood. Here we show epigenetic reprogramming shapes the cellular landscape of schwannomas, the most common tumors of the peripheral nervous system. We find schwannomas are comprised of 2 molecular groups that are distinguished by activation of neural crest or nerve injury pathways that specify tumor cell states and the architecture of the tumor immune microenvironment. Moreover, we find radiotherapy is sufficient for interconversion of neural crest schwannomas to immune-enriched schwannomas through epigenetic and metabolic reprogramming. To define mechanisms underlying schwannoma groups, we develop a technique for simultaneous interrogation of chromatin accessibility and gene expression coupled with genetic and therapeutic perturbations in single-nuclei. Our results elucidate a framework for understanding epigenetic drivers of tumor evolution and establish a paradigm of epigenetic and metabolic reprograming of cancer cells that shapes the immune microenvironment in response to radiotherapy.

Cancer is a heterogeneous disease, and the evolution of cell states and cell types in the tumor microenvironment can influence response to treatment[1–5]. Peripheral nervous system Schwann cells develop from the neural crest[6], a multipotent embryonic cell population characterized by remarkable molecular and functional diversity[7]. Schwannoma tumors have a low burden of somatic mutations that do not change after treatment with ionizing radiation[8,9], but schwannomas that are treated with radiotherapy periodically swell and shrink for many years[10,11]. In clinical practice, symptomatic schwannoma oscillations are treated with empiric immunosuppressive corticosteroids or

surgical decompression, and preoperative schwannoma growth is associated with immune cell infiltration[12,13].

Here we test the hypothesis that epigenetic mechanisms shape schwannoma cell states and the immune microenvironment during schwannoma responses to radiotherapy. To do so, we interrogate human schwannomas, primary patient-derived schwannoma cells, and Schwann and schwannoma cell lines using bulk and single-cell bioinformatics, functional genomic, proteomic, metabolomic, and mechanistic approaches. We find schwannomas are comprised of 2 molecular groups that are distinguished by tumor and immune cell

types and can arise de novo (Fig. 1) but undergo epigenetic inter-conversion in response to radiotherapy (Fig. 2). Genome-wide CRISPR interference (CRISPRi) screening[14,15] identifies epigenetic regulators driving schwannoma cell reprogramming and immune cell infiltration in response to ionizing radiation (Fig. 3), and a technique integrating single-nuclei ATAC, RNA, and CRISPRi perturbation sequencing elucidates concordant chromatin accessibility, transcription factor activity, and gene expression programs underlying schwannoma cell state evolution that are conserved in human tumors (Fig. 4). In sum, these data shed light on the molecular landscape of schwannomas and reveal epigenetic mechanisms underlying tumor heterogeneity and response to radiotherapy (Fig. 5).

## Results

### Schwannomas are comprised of neural crest and immune-enriched molecular groups

DNA methylation profiling provides a robust classification of nervous system tumors and identifies biological drivers of tumorigenesis and treatment response[16,17]. To define a molecular architecture for mechanistic interrogation of schwannomas, DNA methylation profiling was performed on 66 sporadic vestibular schwannomas from the University of California San Francisco (Fig. 1a, Supplementary Figs. 1 and 2, and Supplementary Data 1–3). Unsupervised hierarchical clustering and K-means consensus clustering of differentially methylated DNA probes revealed 2 molecular groups (Fig. 1a and Supplementary Fig. 1a). Differentially methylated DNA probes across schwannoma molecular groups were enriched at genes involved in nervous system development or immunologic signaling (Fig. 1a and Supplementary Data 2). Unsupervised hierarchical clustering of 125 sporadic spinal or vestibular schwannomas from an independent institution also identified 2 molecular groups that were distinguished by differential DNA methylation of nervous system development or immune genes (Supplementary Fig. 1b)[8]. Schwannoma clustering using DNA methylation probes overlapping with active enhancers during neural crest development recapitulated 2 molecular groups of tumors (Supplementary Fig. 2a, b)[18]. RNA sequencing and differential expression analysis on 11 neural crest and 13 immune-enriched schwannomas integrated with DNA methylation profiling revealed enrichment and hypomethylation of either neural crest genes and Hedgehog target genes or immune genes and apolipoprotein genes across the 2 molecular groups of schwannomas (Supplementary Fig. 2c, d and Supplementary Data 3)[19]. Immunofluorescence on 49 schwannomas showed the Schwann cell differentiation marker SOX10 was expressed in both molecular groups but was enriched in neural crest compared to immune-enriched tumors (Supplementary Fig. 2e). Histologic assessment of 66 schwannomas showed immune-enriched tumors were distinguished by greater macrophage and lymphocyte infiltration, hyalinized vessels, and coagulative necrosis compared to neural crest schwannomas (Supplementary Fig. 2f).

To define cell types comprising schwannoma molecular groups, single-nuclei or single-cell RNA sequencing was performed on 4 neural crest schwannomas and 5 immune-enriched schwannomas (Fig. 1b–d and Supplementary Figs. 3 and 4). Integration of 10,628 single-nuclei or single-cell transcriptomes using Harmony[20] revealed 13 cell clusters in uniform manifold approximation and projection (UMAP) space (Fig. 1b). The expression of Schwann cell differentiation markers S100B and SOX10 was used to identify 7 clusters of schwannoma cells and 6 clusters of microenvironment cells (Supplementary Fig. 3a, b). Schwannoma and microenvironment cell types were defined using differentially expressed marker genes (Supplementary Fig. 3c and Supplementary Data 4), unbiased gene expression signatures (Supplementary Fig. 4), and genes associated with active enhancers during demyelinating nerve injury[21] (Supplementary Fig. 3d). Schwannoma cell types reflected different stages of nerve injury and regeneration (Fig. 1b), which collectively involves axon injury, progenitor cell proliferation, myelin remodeling, vascular remodeling, complement activation, immune recruitment, and neural regeneration[22]. Comparison of tumor cell types in immune-enriched versus neural crest schwannomas showed enrichment of (1) immune-like schwannoma cells expressing BCL1 and CD74, a regulator of macrophage migration, axon repair, and survival of neural progenitor cells;[23,24] (2) suppression of vascular remodeling schwannoma cells expressing ADGRB3, a regulator of angiogenesis and cell proliferation[25–27], and (3) suppression of axon injury schwannoma cells expressing NRXN3, a regulator of synapse function[28] (Fig. 1c, d). In support of these findings, immunohistochemistry and immunofluorescence revealed schwannoma cells expressing BCL1 were more common in immune-enriched compared to neural crest schwannomas (Supplementary Fig. 3e, f).

To complement our transcriptomics and better define immune cell types in the schwannoma microenvironment, mass cytometry by time-of-flight (CyTOF) analysis was performed on 375,355 cells from 3 neural crest schwannomas and 3 immune-enriched schwannomas (Fig. 1e–h, Supplementary Figs. 5–9, and Supplementary Data 5). CD45+ immune cells were visualized in toto using Scaffold v0.1 maps[29] (Fig. 1e and Supplementary Fig. 5a). Proliferating immune cells adjacent to spike-in landmark myeloid or lymphoid clusters were greater in immune-enriched compared to neural crest schwannomas, including memory and effector CD4 and CD8 T cells, γδT cells, NK cells, macrophages, and conventional and plasmacytoid dendritic cells (Fig. 1e). UMAP and PhenoGraph clustering were used to define myeloid and lymphoid immune cell phenotypes[30] (Fig. 1f–h, Supplementary Fig. 5b, and Supplementary Figs. 6–9). In comparison to neural crest schwannomas, myeloid cells in immune-enriched schwannomas accumulated more macrophages lacking CD206 and with lower CD163 expression, a pattern associated with anti-tumor activity[31–33]. PD1 + TEMRA CD8 T cells, which are associated with differentiated effector function and tumor control[34,35], were also enriched in immune-enriched compared to neural crest schwannomas. In support of these phenotypes, immunohistochemistry for CD68, CD3, and PD1 was greater in immune-enriched compared to neural crest schwannomas (Supplementary Fig. 5c).

To determine if schwannoma molecular groups could be identified preoperatively, we analyzed magnetic resonance (MR) imaging and clinical features for 31 neural crest and 35 immune-enriched schwannomas (Fig. 1i, j, Supplementary Fig. 10, and Supplementary Data 1). Neural crest schwannomas presented as solid masses with restricted MR diffusion compared to immune-enriched tumors (48% versus 15%, $p = 0.005$, Fisher's exact test) (Fig. 1i). Immune-enriched schwannomas presented with communicating hydrocephalus (38% versus 15%, $p = 0.04$) and cystic changes (53% versus 27%, $p = 0.03$) and had residual tumor adherent to the brainstem after resection (56% versus 21%, $p = 0.02$, Fisher's exact tests) even after adjusting for tumor size (OR 5.3, 95% CI: 1.4–24.1, $p = 0.02$). Univariate and recursive partitioning analyses of MR imaging and clinical features showed that prior schwannoma radiotherapy, cystic changes, and the absence of reduced diffusion were specific to immune-enriched schwannomas (Supplementary Fig. 10a). Logistic regression using these features yielded a model with 79% sensitivity and 77% specificity in distinguishing neural crest and immune-enriched schwannomas (AUC 0.79) (Fig. 1j and Supplementary Fig. 10b). Analysis of clinical outcomes showed local freedom from recurrence was equivalent across schwannoma molecular groups (Supplementary Fig. 10c).

### Radiotherapy is sufficient for epigenetic reprogramming of neural crest to immune-enriched schwannoma

Patients with prior schwannoma radiotherapy (42.9% versus 6.5%, $p = 0.0007$, Fisher's exact test) were more likely to have immune-enriched schwannomas than neural crest schwannomas (Fig. 1a, Supplementary Figs. 11 and 12, and Supplementary Data 1). Analysis of DNA methylation profiles across clinical features revealed a disproportionately higher number of differentially methylated DNA

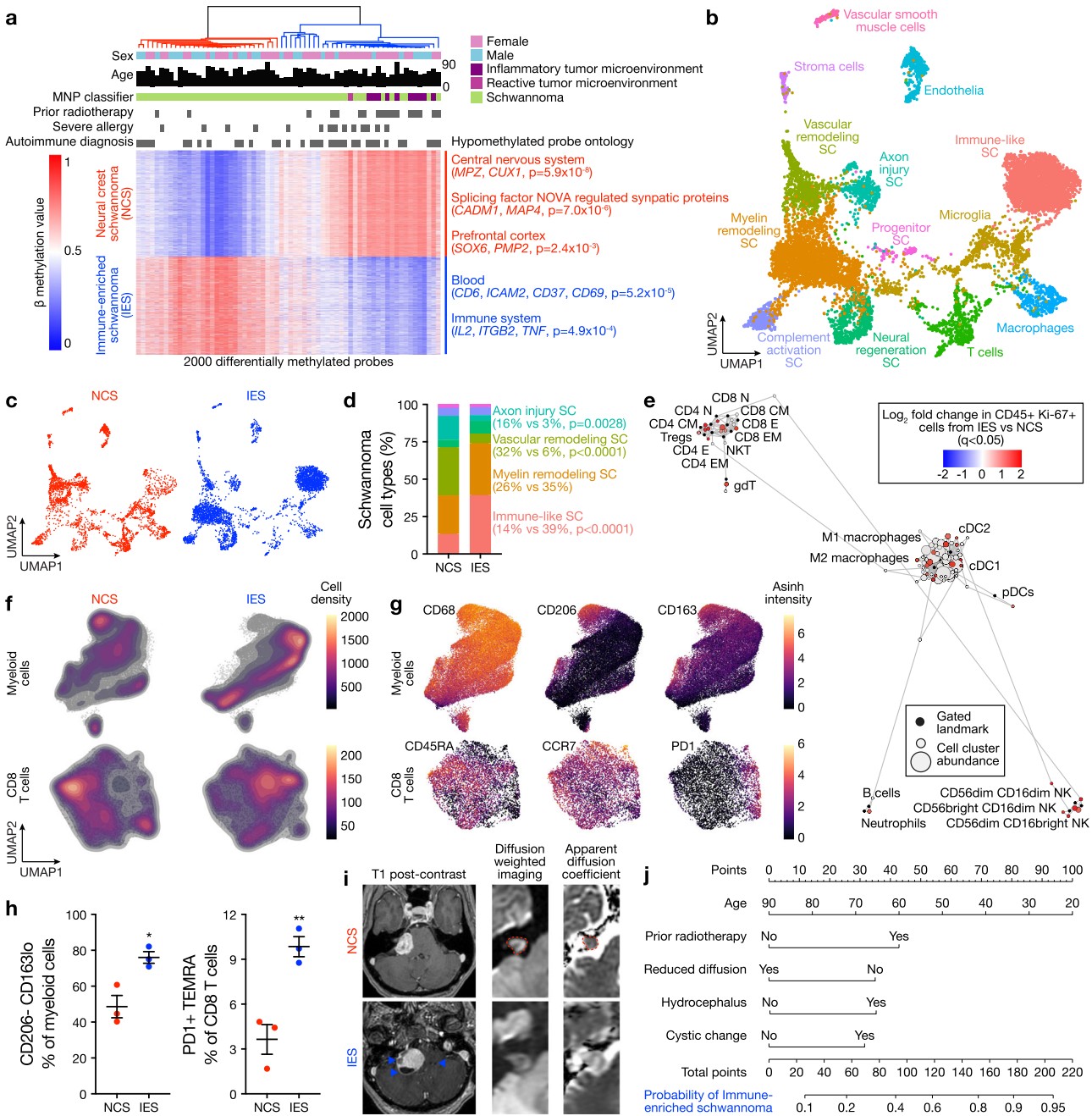

**Fig. 1 | Schwannomas are comprised of neural crest and immune-enriched molecular groups. a** Hierarchical clustering of the top 2000 most differentially methylated probes from 66 vestibular schwannomas. Significant gene ontology terms corresponding to hypomethylated probes in neural crest schwannomas (NCS) or immune-enriched schwannomas (IES), clinical metadata, and the molecular neuropathology (MNP) DNA methylation classification of central nervous system tumors[16] are shown. **b** Integrated UMAP of 10,628 transcriptomes from harmonized schwannoma single-nuclei (n = 6) or single-cell (n = 3) RNA sequencing showing schwannoma cell (SC) types and tumor microenvironment cell types. **c** UMAPs from **b** with individual transcriptomes split according to a molecular group of origin. **d** Relative composition of SC types according to a molecular group of origin, colored as in (**b**). **e** Scaffold plot comprised of 375,355 Ki67+ immune cells from NCS (n = 3) or IES (n = 3) analyzed using mass cytometry time-of-flight (CyTOF). Manually gated landmark immune cell populations (black) are annotated. Schwannoma immune cell cluster is colored when the proportion of cells is statistically

different between IES (positive) and NCS (negative). **f** CyTOF UMAPs of CD45+ immune cells with overlaid density plots for manually gated myeloid cells (top, 40,000 cells) or CD8 T cells (bottom, 6632 cells) in NCS (left) or IES (right). **g** UMAP feature plots of marker genes used to define myeloid or CD8 T cells in (**f**). **h** CyTOF proportion of schwannoma myeloid cells (left) or CD8 T cells (right) corresponding to M1 macrophages or PD1 + TEMRA CD8 T cells, respectively, in NCS (n = 3 independent tumors) or IES (n = 3 independent tumors). Lines represent means, and error bars represent the standard error of means (two-sided Student's t-tests, *p = 0.0174, **p = 0.0065). **i** Representative preoperative magnetic resonance imaging of 66 vestibular schwannomas using T1 post-contrast, T2 diffusion-weighted, or apparent diffusion coefficient (ADC) sequences reveals NCS present as solid masses with reduced diffusion (dotted line) and IES present with cystic changes (arrows) and hydrocephalus. **j** Nomogram for schwannoma immune enrichment and molecular grouping based on non-invasive clinical and magnetic resonance imaging features. Source data are provided as a Source Data file.

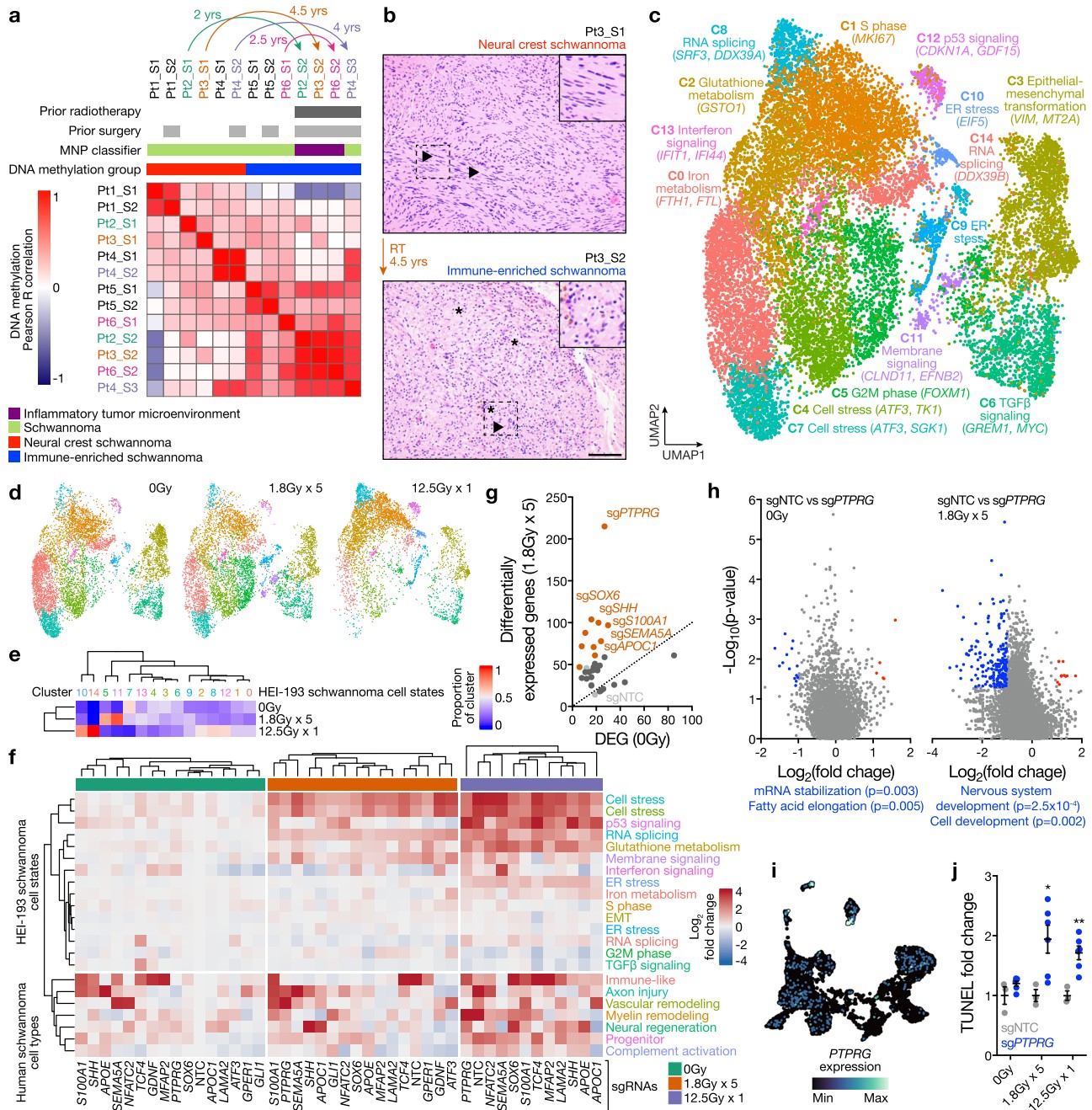

**Fig. 2 | Radiotherapy is sufficient for epigenetic reprogramming of neural crest to immune-enriched schwannoma. a** Pairwise Pearson correlation coefficients grouped by hierarchical clustering of DNA methylation profiles from the patient (Pt) matched primary and recurrent schwannomas (*n* = 13). Arrows represent the reprogramming of primary NCS to IES at recurrence. **b** H&E-stained sections of a patient-matched primary NCS and recurrent IES, showing Verocay bodies (top arrows) and abundant lymphocytes (bottom arrows) with foamy and hemosiderin-filled macrophages (asterisks). Scale bar, 100 μm. Similar findings were observed in two additional matched pairs. **c** UMAP of 38,754 transcriptomes from single-cell RNA sequencing of HEI-193 cells after treatment with 0Gy, 1.8 Gy × 5, or 12.5 Gy × 1 (*n* = 3 independent replicates per condition) of radiotherapy revealing 15 distinct schwannoma cell states. **d** UMAPs from **c** with individual transcriptomes split according to triplicate treatment conditions. **e** Relative composition of cell states from **c** according to treatment conditions. **f** Perturb-seq gene-set expression heatmap of 3546 pseudobulked transcriptomes from HEI-193 cells following sgRNA perturbations (columns) across treatment conditions. Expression values are normalized to cells harboring non-targeting control sgRNAs (sgNTC) with 0 Gy.

**g** Number of differentially expressed genes (DEG) from schwannoma cell Perturb-seq with radiotherapy (y-axis) versus 0Gy (x-axis). sgRNA perturbations with ≥40 DEG after radiotherapy compared to control are orange. sgRNA not meeting this threshold are dark gray. sgNTCs are light gray. **h** Differential gene expression analysis from *PTPRG* perturbation compared to sgNTC in 0 Gy (left) versus 1.8 Gy × 5 conditions (right). Significant positive (red) or negative (blue) gene expression changes are colored (*p* < 0.05, |log₂(fold change)| > 1), corresponding to gene ontology (two-sided Fisher's exact test without adjustments for multiple comparisons). **i** Feature plot of integrated UMAP from harmonized schwannoma single transcriptomes (Fig. 1b) showing *PTPRG* expression in schwannoma cells. **j** TUNEL staining for apoptosis in HEI-193 cells following CRISPRi suppression of *PTPRG* versus sgNTC after treatment with 0 Gy (*n* = 3 independent cultures), 1.8 Gy × 5 (*n* = 3 independent cultures across 2 sgRNAs each), or 12.5 Gy × 1 (*n* = 3 independent cultures across 2 sgRNAs each) of radiotherapy. Fold changes are normalized to sgNTC in each treatment condition. Lines represent means, and error bars represent the standard error of means (two-sided Student's t-tests, *\*p* = 0.030, *\*\*p* = 0.0044). Source data are provided as a Source Data file.

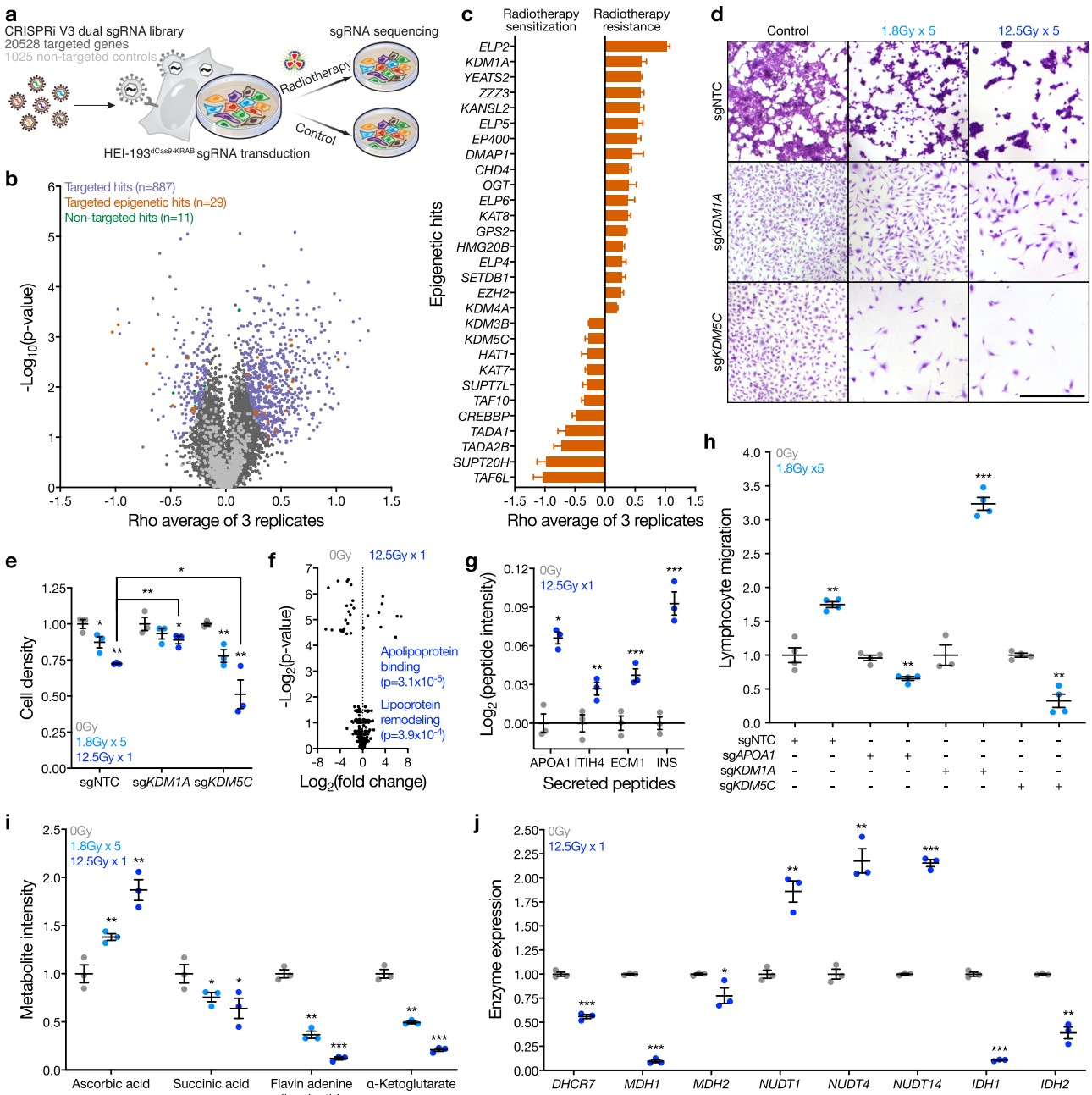

**Fig. 3 | Epigenetic regulators reprogram schwannoma cells and drive immune cell infiltration in response to radiotherapy. a** Experimental workflow for genome-wide CRISPRi screens using dual sgRNA libraries comprised of 20,528 targeted sgRNAs and 1025 non-targeted control sgRNAs (sgNTC). Libraries were transduced into HEI-193 cells that were subsequently treated with 0 Gy or 1.8 Gy × 5 radiotherapy ($n = 3$ per condition). sgRNA barcodes were sequenced and quantified as proxies for cell enrichment or depletion. **b** CRISPRi screen results showing the average rho log₂(sgRNA in radiotherapy conditions/sgRNA in control conditions) and two-sided Student's $t$-test $p$ values across three replicates. On-target hit genes (purple), epigenetic regulator hit genes (orange), and sgNTCs called hits (green) at a false discovery rate of 1% are shown. **c** Rho phenotypes of epigenetic regulator CRISPRi screen hits genes from (**b**). $n = 3$ replicate screens. **d** Crystal violet staining of HEI-193 following CRISPRi of *KDM1A* or *KDM5C* compared to sgNTC after treatment with 0 Gy, 1.8 Gy × 5, or 12.5 Gy × 1 of radiotherapy. Scale bar, 100 µm. **e** Quantification of cell density from (**d**) ($n = 3$ independent cultures, two-sided Student's $t$-test). **f** Volcano plot of 425 peptides identified using proteomic mass spectrometry of conditioned media from triplicate HEI-193 cultures after

radiotherapy or control treatment. Significant gene ontology terms of enriched peptides after radiotherapy conditions annotated (two-sided Fisher's exact test). **g** Proteomic mass spectrometry parallel reaction monitoring targeted assay validating secreted peptide enrichment in conditioned media from HEI-193 after radiotherapy ($n = 3$ independent cultures, two-sided Student's $t$-test) as in (**f**). **h** Transwell primary human peripheral blood lymphocyte migration assays using conditioned media from HEI-193 ($n = 3$ independent cultures, two-sided Student's $t$-test) following CRISPRi suppression of *APOA1*, *KDM1A*, or *KDM5C* ± radiotherapy as a chemoattractant. **i** Targeted metabolite mass spectrometry of HEI-193 after treatment with 0 Gy, 1.8 Gy × 5, or 12.5 Gy × 1 of radiotherapy ($n = 3$ independent cultures per condition, two-sided Student's $t$-test). Fold changes normalized to 0 Gy treatment for each metabolite. **j** Metabolic enzymes gene expression changes from bulk RNA sequencing of HEI-193 cells ± radiotherapy ($n = 3$ independent cultures) (Supplementary Data 6). Fold changes normalized to 0 Gy treatment for each gene. Lines represent means, and error bars represent standard error of means (two-sided Student's $t$-tests, *$p \le 0.05$, **$p \le 0.01$, ***$p \le 0.0001$). Source data are provided as a Source Data file.

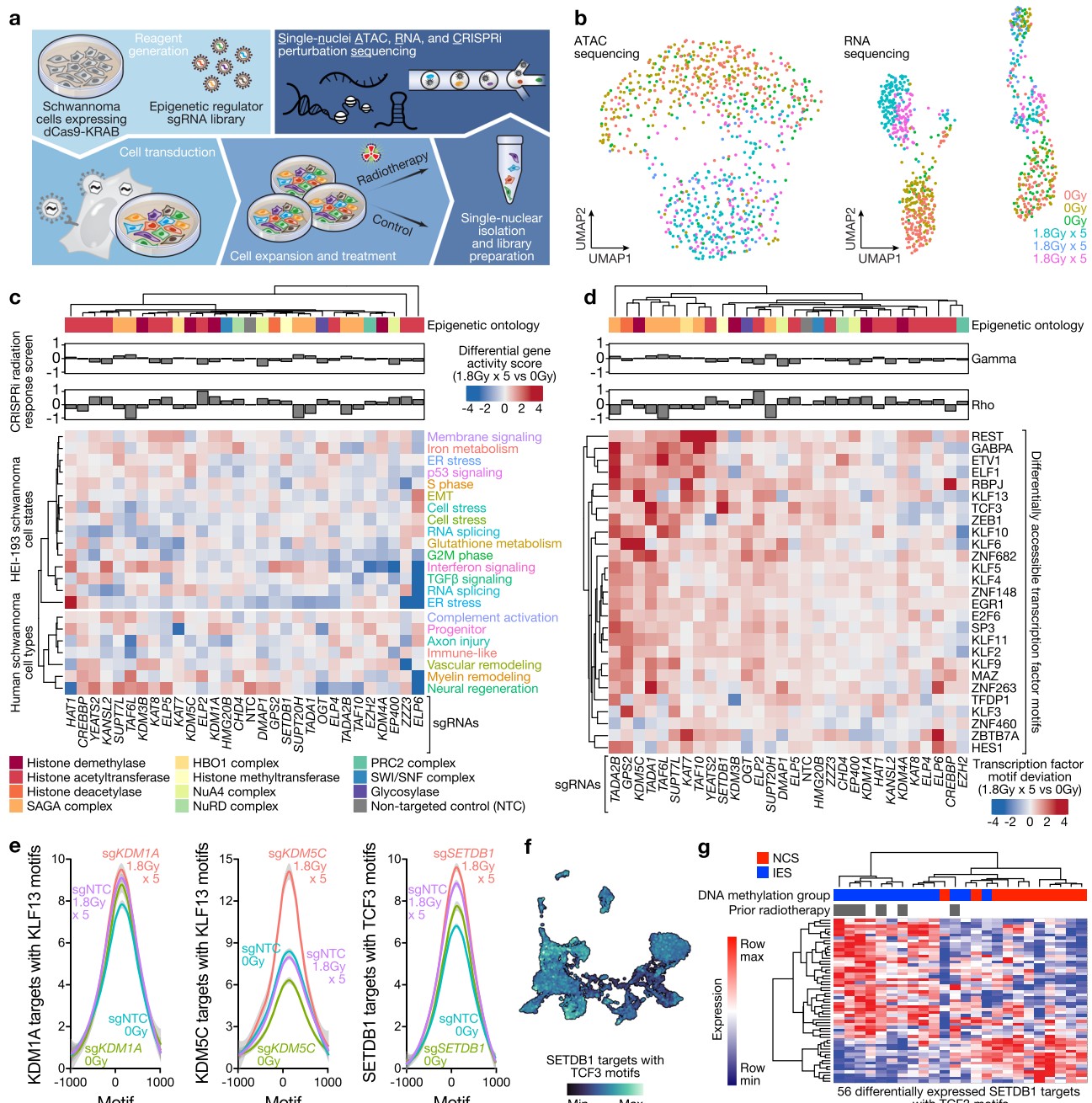

**Fig. 4 | Single-nuclei ATAC, RNA, and CRISPRi perturbation sequencing identify integrated genomic mechanisms driving schwannoma cell state evolution. a** Experimental workflow for single-nuclei ATAC, RNA, and CRISPRi perturbation sequencing (snARC-seq). Triplicate HEI-193 cultures were transduced with sgRNA libraries targeting 29 epigenetic regulators driving radiotherapy responses from genome-wide CRISPRi screens (Fig. 3c) and treated with 0 Gy or 1.8 Gy × 5 of radiotherapy prior to isolation of single-nuclei for sequencing. sgRNA identities were recovered from CROP-seq tags in single-nuclei RNA sequencing data. **b** UMAPs of ATAC (left) or RNA (right) sequencing of 855 single nuclei passing snARC-seq quality control from triplicate control or radiotherapy conditions (Supplementary Fig. 18). **c** Hierarchical clustering of differential gene activity scores between radiotherapy and control conditions for each snARC-seq perturbation (columns). Gene activity modules (rows) were derived from HEI-193 schwannoma cell states ± radiotherapy (Fig. 2c) or from human schwannoma cell types (Fig. 1b).

Gene ontology of perturbed epigenetic regulators and CRISPRi screen growth (gamma) or radiation response (rho) phenotypes from genome-wide CRISPRi screens (Fig. 3c) are shown. **d** Hierarchical clustering of differential ChromVAR transcription factor motif deviations between radiotherapy and control conditions for each snARC-seq perturbation (columns). **e** Average profile plots of normalized ATAC signal at *KLF13* or *TCF3* motifs with ENCODE ChIP-seq peak annotations and differential accessibility following snARC-seq perturbation of *KDM1A*, *KDM5C*, or *SETDB1*. **f** Feature plot of integrated UMAP from harmonized schwannoma single-nuclei and single-cell RNA sequencing (Fig. 1b) showing genes near TCF3 motifs that are differentially accessible following *SETDB1* snARC-seq perturbation. **g** Hierarchical clustering of human schwannoma RNA sequencing profiles using 56 differentially expressed SETDB1 targets with TCF3 motifs showing separation of NCS and IES molecular groups of schwannomas. Source data are provided as a Source Data file.

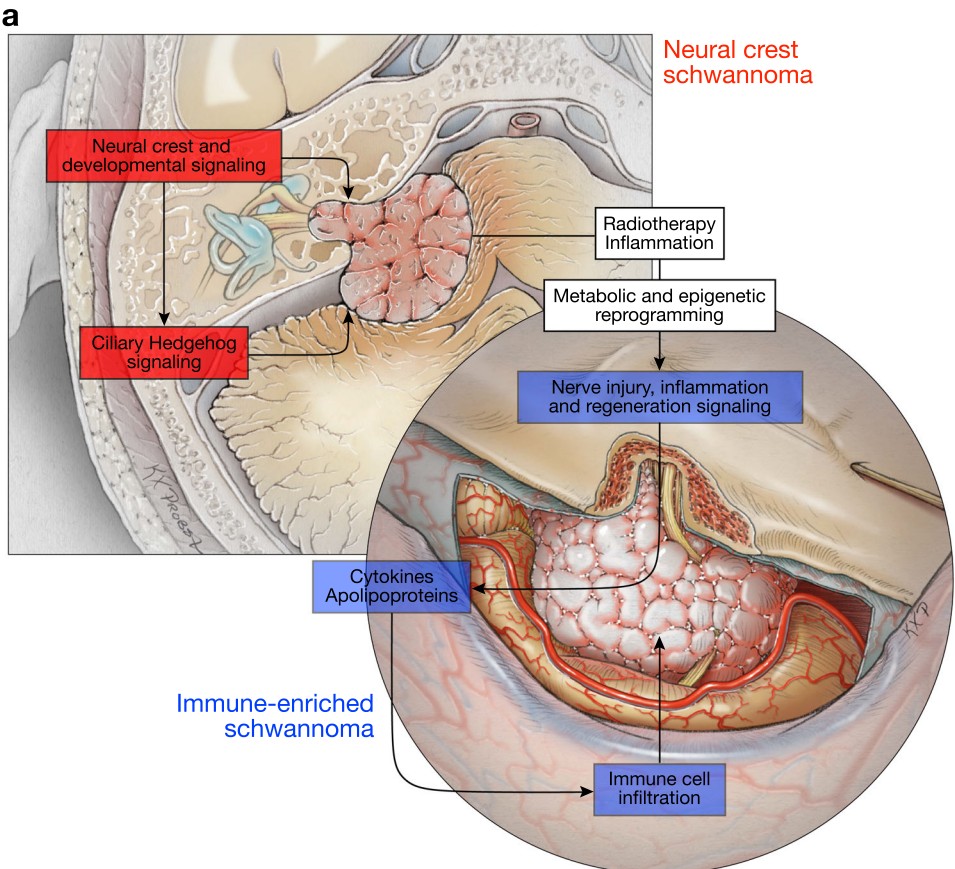

**Fig. 5 | An integrated model of schwannoma tumorigenesis and epigenetic reprogramming in response to treatment.** Two molecular groups of schwannoma, neural crest schwannoma and immune-enriched schwannoma, are driven by distinct mechanisms. Radiotherapy can induce immune-enriched schwannoma through metabolic and epigenetic reprogramming.

probes in schwannomas could be attributed to prior radiotherapy (Supplementary Fig. 11a). Hierarchical clustering of differentially methylated DNA probes distinguishing schwannomas with prior radiotherapy showed hypomethylation of immune genes and hypermethylation of neuronal progenitor maintenance genes[36] (Supplementary Fig. 11b). Comparison of DNA methylation profiles between paired primary and recurrent schwannomas from 6 patients (Supplementary Data 1) revealed epigenetic interconversion of recurrent tumors compared to cognate primary tumors in every instance where radiotherapy was delivered between serial surgeries (Fig. 2a). All recurrent schwannomas with prior radiotherapy from this matched set were immune-enriched and clustered together with pairwise Pearson correlation coefficients that were indistinguishable from randomly-selected unpaired schwannomas with prior radiotherapy (Supplementary Fig. 11c). Moreover, primary neural crest schwannomas with classic histology were infiltrated by greater numbers of lymphocytes and macrophages that persisted for at least 4 years after radiotherapy (Fig. 2b and Supplementary Fig. 12).

RNA sequencing and differential expression analysis showed Hedgehog target genes were enriched in schwannomas without prior radiotherapy (Supplementary Fig. 11d and Supplementary Data 6). Thus, we hypothesized Hedgehog signaling may underlie neural crest schwannomas. Hedgehog signals are transduced through primary cilia[37]. Immunofluorescence revealed schwannoma cell cilia were longer in neural crest compared to immune-enriched schwannomas (Supplementary Fig. 13a), and single-cell and single-nuclei RNA sequencing validated Hedgehog target gene expression in schwannoma cells from human tumors (Supplementary Fig. 13b). Primary cilia were also identified on cultured human Schwann cells, which

accumulated Smoothened in cilia and activated the Hedgehog transcriptional program in response to recombinant Sonic Hedgehog (SHH) (Supplementary Fig. 13c–e). Schwann cell proliferation was blocked by the Smoothened antagonist vismodegib or radiotherapy (Supplementary Fig. 13f, g), and radiotherapy also reduced ciliary length, attenuated Smoothened accumulation in primary cilia, and blocked Hedgehog target gene expression (Supplementary Fig. 13h–j). To study these mechanisms in human schwannoma cells, we used the HEI-193 cell line, which encodes heterozygous loss of *NF2* on chromosome 22q and a distal splice site mutation in the remaining *NF2* allele[38,39]. CRISPRi suppression of *NF2* in HEI-193 cells stably expressing dCas9-KRAB[14,40] promoted cell proliferation compared to HEI-193 cells expressing non-targeted control sgRNAs (sgNTC) (Supplementary Fig. 13k, l), suggesting *NF2* retains tumor suppressor functionality in this cell line. Consistent with results from human Schwann cells (Supplementary Fig. 13h–j), radiotherapy blocked HEI-193 Hedgehog target gene expression (Supplementary Fig. 13m). In human tumors, the intraflagellar transport gene *IFT88*, which is necessary for assembly of primary cilia[37] and regulates the G1-S transition[41,42], was hypermethylated and suppressed in immune-enriched compared to neural crest schwannomas (Supplementary Fig. 13n, o).

To more broadly define schwannoma cell responses to ionizing radiation, HEI-193 schwannoma cells were treated with fractionated (1.8 Gy x 5 fractions) or hypofractionated (12.5 Gy x 1 fraction) radiotherapy, both of which are used to treat human schwannomas[6]. Radiotherapy blocked the growth of schwannoma cells (Supplementary Fig. 14a, b), and qPCR assessment of genes distinguishing molecular groups of human schwannomas (Fig. 1) showed radiotherapy induced schwannoma cell expression of inflammatory

apolipoproteins[43,44], immediate early genes that drive cell fate decisions[45], and *SOX6*, an inhibitor of oligodendrocyte cell differentiation[46] (Supplementary Fig. 14c). To determine if these genes promoted schwannoma cell survival, *APOD* or *SOX6* were suppressed in HEI-193 cells using CRISPRi, which each attenuated schwannoma cell proliferation compared to cells expressing sgNTC (Supplementary Fig. 14d, e). DNA methylation profiling and RNA sequencing revealed hypomethylation and enriched expression of immune and apolipoprotein genes in surviving HEI-193 cells after radiotherapy (Supplementary Fig. 14f, g). To determine if radiotherapy reprograms schwannoma cell states, single cell RNA-sequencing was performed on HEI-193 cells after control, fractionated radiotherapy, or hypofractionated radiotherapy treatments (Supplementary Fig. 14h). Clustering of 39,569 single-cell transcriptomes revealed 15 cell states in UMAP space (Fig. 2c). Schwannoma cell states were defined using differentially expressed marker genes (Supplementary Fig. 14i and Supplementary Data 7), revealing activation of endoplasmic reticulum (ER) stress, interferon signaling, membrane signaling, and p53 signaling in response to radiotherapy (Fig. 2c–e).

To define how schwannoma cell states (Fig. 2c) or cell types (Fig. 1b) respond to radiotherapy, we used Perturb-seq, a functional genomic approach coupling CRISPRi screening with transcriptomic phenotypes in single cells[47–49]. Perturb-seq suppression of 15 genes distinguishing molecular groups of human schwannomas or schwannoma cell states (Fig. 1) was performed in HEI-193 cells with control, fractionated radiotherapy, or hypofractionated radiotherapy treatments (Supplementary Data 8). Integration of single-cell transcriptomes revealed intercellular heterogeneity that was dependent on radiotherapy dose and the cell cycle (Supplementary Fig. 15a–e). We then asked whether Perturb-seq gene suppression affected gene modules distinguishing the 15 schwannoma cell states from single-cell RNA sequencing of schwannoma cells (Fig. 2c), or the 7 schwannoma cell types reflecting different stages of nerve injury and regeneration from integrated single-nuclei and single-cell RNA sequencing of human schwannomas (Fig. 1b). Radiotherapy dose was the primary determinant of cell stress, p53 pathway, RNA splicing, and glutathione pathway activation, but genetic perturbations caused heterogenous changes in gene module expression with and without ionizing radiation (Fig. 2f). To identify perturbations selectively disrupting gene expression programs in combination with radiotherapy, differential expression analysis was performed on single cells expressing targeted versus non-targeted sgRNAs with or without radiotherapy (Supplementary Data 9). Suppression of *SHH*, *SOX6*, or the receptor tyrosine phosphatase *PTPRG*, which regulates neural developmental signaling[50], exhibited greater numbers of differentially expressed genes after radiotherapy compared to control treatment (Fig. 2g, h and Supplementary Fig. 15f). Integrated single-nuclei and single-cell RNA sequencing showed *PTPRG* was expressed in schwannoma cells from human tumors (Fig. 2i), Perturb-seq suppression of *PTPRG* in schwannoma cells inhibited genes underlying nervous system development only in combination with radiotherapy (Fig. 2h), and CRISPRi suppression of *PTPRG* enhanced schwannoma cell apoptosis with radiotherapy (Fig. 2j and Supplementary Fig. 15g, h). These data reveal polygenic mechanisms underlie reprograming of neural crest to immune-enriched schwannoma and suggest a role for *PTPRG* in schwannoma response to radiotherapy.

## Epigenetic regulators reprogram schwannoma cells and drive immune cell infiltration in response to radiotherapy

To more broadly define genomic drivers specifying schwannoma cell states in response to ionizing radiation, triplicate genome-wide CRISPRi screens were performed using dual sgRNA libraries comprised of 20,528 targeted sgRNAs and 1025 sgNTCs[15] (Fig. 3a, Supplementary Fig. 16, 17, and Supplementary Data 10). The effect of genetic perturbations on HEI-193 schwannoma cell growth ± radiotherapy was defined by quantifying the relative DNA abundance of all sgRNAs in

each sample using sequencing of integrated barcodes (Supplementary Data 10). Gene set enrichment analysis of radiotherapy sensitivity screen hits showed expected enrichment of cell cycle and DNA repair, but also revealed significant enrichment of epigenetic genes (Supplementary Fig. 16a). Examination of screen hits identified 29 epigenetic regulators associated with radiotherapy resistance or sensitivity phenotypes in schwannoma cells (Fig. 3b, c). In support of these findings, CRISPRi suppression of the histone demethylases *KDM1A* or *KDM5C* validated radiotherapy resistance or sensitivity phenotypes, respectively (Fig. 3d, e and Supplementary Fig. 16b).

Cancer cell death from ionizing radiation leads to acute, transient recruitment of immune cells to the tumor microenvironment[51], but we identified surviving schwannoma cells alongside infiltrating immune cells for many years after treatment of human tumors with radiotherapy (Fig. 2b and Supplementary Fig. 2f, 5c, 12). To determine if schwannoma cell reprogramming contributes to immune cell infiltration of the tumor microenvironment, proteomic mass spectrometry was performed on conditioned media from surviving HEI-193 schwannoma cells after radiotherapy (Supplementary Fig. 17a–c). Ontology analyses of 425 differentially expressed proteins from conditioned media revealed enrichment of apolipoproteins after ionizing radiation (Fig. 3f), and a parallel reaction monitoring targeted assay validated secretion of APOA1 and other chemokines from surviving schwannoma cells (Fig. 3g). Transwell migration assays showed conditioned media from schwannoma cells recruited primary human peripheral blood lymphocytes after radiotherapy (Supplementary Fig. 16c), and conditioned media from schwannoma cells with CRISPRi suppression of the radiotherapy resistance hit *KDM1A* enhanced lymphocyte migration (Fig. 3h). In contrast, conditioned media from schwannoma cells with CRISPRi suppression of *APOA1* or the radiotherapy sensitivity hit *KDM5C* inhibited lymphocyte migration (Fig. 3h and Supplementary Fig. 16b). Thus, epigenetic mechanisms in schwannoma cells contribute to immune cell infiltration in response to radiotherapy.

To determine if altered expression of epigenetic regulators mediates schwannoma responses to radiotherapy, we analyzed the 29 epigenetic hits from our genome-wide CRISPRi screen (Fig. 3b, c) across schwannoma cell types (Fig. 1b) and schwannoma cell states (Fig. 2c). Surprisingly, no epigenetic regulators were differentially expressed across either context, either with or without radiotherapy (Supplementary Data 4,7). Epigenetic regulator activity is dependent on metabolite cofactors that covalently modify histone subunits[52]. Thus, we hypothesized schwannoma cell metabolites may be altered in response to radiotherapy. To test this, we performed targeted metabolite profiling of HEI-193 cells after control, fractionated radiotherapy, or hypofractionated radiotherapy treatments using liquid chromatography mass spectrometry. Radiotherapy suppressed the KDM5C cofactor α-ketoglutarate and succinic acid, a biproduct of α-ketoglutarate metabolism (Fig. 3i). α-ketoglutarate was also suppressed by radiotherapy in primary schwannoma cells from 7 patients (Supplementary Fig. 17d). Moreover, radiotherapy suppressed the KDM1A cofactor flavin adenine dinucleotide (FAD) and increased the KDM5C cofactor ascorbic acid (Fig. 3i). RNA sequencing showed *IDH1/2* and *MDH1/2*, which produce α-ketoglutarate or succinic acid, respectively, were suppressed following schwannoma cell radiotherapy (Fig. 3j). *DHCR7*, which metabolizes ascorbic acid precursors[53], was also suppressed after radiotherapy, but the nudix hydrolases *NUTD1/4/14*, which degrade FAD, were increased (Fig. 3j). These data suggest altered expression of metabolic enzymes and metabolite cofactors may influence epigenetic regulator activity during schwannoma radiotherapy responses.

## Single-nuclei ATAC, RNA, and CRISPRi perturbation sequencing identifies integrated genomic mechanisms driving schwannoma cell state evolution

To define how epigenetic regulators shape chromatin accessibility and gene expression in schwannoma cells during tumor evolution, we

developed a technique for simultaneous interrogation of chromatin accessibility and gene expression coupled with genetic and therapeutic perturbations in single-nuclei (Fig. 4a). Single-nuclei ATAC, RNA, and CRISPRi perturbation sequencing (snARC-seq) of the 29 epigenetic regulator hits from our genome-wide CRISPRi screen (Fig. 3b, c) was performed in HEI-193 cells with control or radiotherapy treatments (Supplementary Fig. 18a). sgRNA identities were assigned to individual cells using a hypergeometric test[54] (Supplementary Fig. 18b, c). Genome-wide ATAC signals were enriched at transcription start sites and 5' nucleosome free regions[55] (Supplementary Fig. 18d). ATAC and RNA sequencing data exhibited heterogenous distributions in UMAP space[56] (Fig. 4b). Together, these data support successful simultaneous profiling of 3 genomic modalities in single-nuclei with or without radiotherapy (Fig. 4a).

To determine if snARC-seq suppression of chromatin regulators reprograms the epigenetic landscape of schwannoma cells, we quantified gene activity scores for open chromatin regions nearby marker genes distinguishing schwannoma cell states (Fig. 2c) or cell types (Fig. 1b). Differential gene activity scores clustered according to gene ontologies of perturbed epigenetic regulators (e.g., histone demethylases, histone acetyltransferases) and CRISPRi screen phenotypes (radiotherapy sensitivity, negative rho; radiotherapy resistance, positive rho) (Fig. 4c). For instance, snARC-seq suppression of the elongator acetyltransferase complex component *ELP6* attenuated interferon signaling and ER stress schwannoma cell states that were normally activated by radiotherapy (Fig. 2c–e), consistent with the role of the elongator acetyltransferase complex in gene activation and sensitivity to genotoxic stress[57,58].

To elucidate transcription factors underlying changes in gene expression programs with snARC-seq suppression of chromatin regulators, we scored transcription factor motif deviation with or without radiotherapy[59] (Fig. 4d). Restricting analysis to transcription factors in HEI-193 cells revealed heterogenous disruptions in chromatin accessibility that clustered based on CRISPRi screen phenotypes (Fig. 4d). snARC-seq suppression of the radiotherapy sensitivity hit *KDM5C* caused enrichment of open chromatin regions at KLF13 motifs with radiotherapy (Fig. 4e), and KLF13 target genes such as *DDI2* and *PTPA* were simultaneously accessible at their genomic loci and had enriched RNA expression in single-nuclei following snARC-seq suppression of *KDM5C* with radiotherapy (Supplementary Fig. 18e). *ITIH4*, a secreted protein that was enriched in conditioned media from schwannoma cells surviving radiotherapy (Fig. 3g), displayed closed chromatin and RNA suppression following snARC-seq suppression of *KDM5C* with radiotherapy (Supplementary Fig. 18e). Concordant changes in KLF13 gene activity were not observed following snARC-seq suppression of the radiotherapy resistance hit *KDM1A* with radiotherapy (Fig. 4e), consistent with opposing phenotypes of *KDM1A* and *KDM5C* in mediating schwannoma radiotherapy responses (Fig. 3c). snARC-seq suppression of the histone methyltransferase *SETDB1* increased accessibility at TCF3 motifs and expression of TCF3 target genes (Fig. 4e) such as *RNF217*, *FOXO1*, and *RAD23B* with radiotherapy (Supplementary Fig. 18e). Integrated single-nuclei and single-cell RNA sequencing showed TCF3 target genes were expressed in schwannoma cells from human tumors (Fig. 4f), and schwannoma clustering restricted to RNA sequencing of TCF3 target genes recapitulated neural crest and immune-enriched molecular groups of schwannomas (Fig. 4g). Thus, our snARC-seq technique for simultaneous interrogation of chromatin accessibility and gene expression coupled with genetic and therapeutic perturbations in single-nuclei reveals epigenetic dependances underlying radiotherapy responses in schwannomas cells that are conserved across molecular groups of human tumors.

## Discussion

Mutational and non-mutational mechanisms are critical for cancer evolution[5,60]. Schwannomas are the most common tumors of the peripheral nervous system and have a remarkably low burden of somatic mutations that is not increased by radiotherapy[8,9,61]. Thus, schwannomas represent prototypical tumors for understanding epigenetic mechanisms specifying tumor cell states, tumor heterogeneity, and response to therapy. Our results reveal schwannomas are comprised of neural crest and immune-enriched molecular groups that are distinguished by schwannoma and immune cell types, and that radiotherapy is sufficient for epigenetic reprogramming of neural crest schwannomas to immune-enriched schwannomas (Fig. 5). By integrating multiplatform profiling of human tumors with multiomic functional genomic approaches, we show interconversion of schwannoma molecular groups, cell types, and cell states in response to radiotherapy, a treatment that is used for half of cancer patients worldwide[62]. Future studies will be needed to determine if the epigenetic mechanisms we report are relevant to other treatment modalities or tumorigenesis. Intriguingly, patients in our study with autoimmune diseases and severe allergies were more liked to have immune-enriched schwannomas than neural crest schwannomas (25.7% vs 0%, p = 0.0025, Fischer's exact test) (Fig. 1a and Supplementary Data 1), suggesting systemic factors may also influence the epigenetic architecture of cancer. To that end, we establish a framework for investigating how tumor evolution and responses to treatment are modulated by epigenetic reprogramming. This paradigm is bolstered by an innovative method for simultaneous profiling of epigenetic and transcriptional cell states coupled with genetic and therapeutic perturbations in single-nuclei (snARC-seq), a technique that may enable new discoveries in health and disease.

## Methods

### Vestibular schwannomas

This study was approved by the UCSF Institutional Review Board (#10-01318, #18-24633) and complied with all relevant ethical regulations. Patients presenting for resection of sporadic vestibular schwannoma who gave written consent for tumor sampling for research were included in the retrospective discovery cohort or in single-cell RNA sequencing and CyTOF experiments (Supplementary Data 1). Exclusion criteria included a history of neurofibromatosis type 2 and, with the exception of prospectively obtained schwannomas for single-cell RNA sequencing and CyTOF experiments, less than 18 months of magnetic resonance imaging (MRI) follow-up. Vestibular schwannomas used for DNA methylation profiling, bulk RNA sequencing, and single-nucleus RNA sequencing were identified from the UCSF Brain Tumor Center Biorepository & Histology Core. All samples from patients meeting inclusion and exclusion criteria were included, resulting in 75 clinically heterogeneous retrospective and prospective schwannomas from 67 patients who were treated from 2003 to 2020. In an attempt to optimize outcomes, many patients in this cohort were initially treated with hybrid strategies employing primary tumor debulking followed by adjuvant radiotherapy[63,64]. As a result, 6 of these patients had tissue from recurrent as well as primary tumors. However, such strategies are now known to be associated with higher recurrence rates and are not recommended. All cases with available material were reviewed by a board-certified neuropathologist for diagnostic confirmation (MP). Additional clinical variables were analyzed by chart review (SJL, JDB, DRR).

### Annotation of radiologic features

A comprehensive radiologic review of preoperative magnetic resonance imaging studies for the patients comprising the retrospective discovery cohort (Supplementary Data 1) was performed by a board-certified neuroradiologist who was blinded to all clinical and molecular data (JEVM). Anatomic magnetic resonance images including T2, T2 fluid attenuated inversion recovery (FLAIR), and post-contrast T1 weighted images were reviewed. Diffusion weighted imaging (DWI) including derived apparent diffusion coefficient (ADC) maps were also

evaluated. Radiologic features of enhancement pattern (homogeneous or heterogeneous), cystic change, mass effect, peritumoral edema, indistinct brain-tumor interface, lobulated margins, and hydrocephalus were scored in categoric binary fashion and were not found to be significantly different between molecular groups of schwannomas unless stated otherwise. Hydrocephalus was further characterized as non-communicating or communicating. Diffusion characteristics were assessed qualitatively (reduced or facilitated) in vestibular schwannomas for which diffusion weighted imaging was acquired.

## DNA methylation profiling and analysis

Genomic DNA was isolated from schwannomas by mechanical homogenization (Qiagen TissueLyser) followed by the All-Prep Universal Kit for DNA and RNA isolation (Qiagen, #80224). DNA quality was tested by spectrophotometry and clean-up was performed using DNA precipitation as needed. Genome-wide methylation profiles were obtained with the Illumina MethylationEPIC 850K array for each schwannoma. Data preprocessing and normalization was performed using *minfi* version 1.30 in R version 3.6.0[65,66]. Methylation probes with detection significance $p > 0.05$ were excluded from analysis. Normalization was performed using functional normalization[67]. Probes were filtered according to the following criteria: (i) exclusion of probes mapping to sex chromosomes (n = 11,551), (ii) exclusion of probes containing a common single nucleotide polymorphism (SNP) within the targeted CpG site or on an adjacent base pair (n = 24,536), and (iii) exclusion of probes not mapping uniquely to the human reference genome hg19 (n = 9,993). A total of 815,630 probes were retained for further analysis. Methylation β values were calculated as the ratio of methylated probe intensity to the sum of methylated and unmethylated probe intensities and visualized on heatmaps in an absolute scale[68]. Variable probes were identified by ranking all methylation probes by β value variance across all schwannomas. The top 2000 probes by variance were then subjected to consensus clustering using complete linkage hierarchical clustering with 1-Pearson correlation as distance matrix, sampling between 2–6 possible clusters with 20 resampling iterations. Schwannomas belonging to each of the 2 molecular groups by hierarchical clustering were used for differential methylation analysis against methylation β values using the dmpFinder function in *minfi*. The resulting differentially methylated probes with False Discovery Rate (FDR) < 0.05 were ranked by log odds values between the 2 molecular groups, and the top and bottom 1000 differentially methylated probes (total 2000) were used for hierarchical clustering of schwannomas. Gene ontology analysis of probes was performed by extracting the associated gene body annotation from each probe using the Bioconductor package IlluminaHumanMethylationEPICanno.ilm10b2.hg19 and inputting the gene lists grouped based on the results of hierarchical clustering into Enrichr[19,69].

To compare DNA methylation with RNA sequencing expression data, we used all methylation probes corresponding to each gene as annotated in the Illumina MethylationEPIC manifest, which includes transcription start site (TSS), gene body, and 3'/5' UTR probes, in order to interrogate beyond promoter methylation. We ranked the differentially methylated probes from most to least significant according to the FDR and compared the most significant probe for each gene to the expression fold change for each differentially expressed gene in RNA sequencing analysis.

Random forest classification of global methylation profiles was performed to compare each methylation profile with a broader set of CNS malignancies as previously described[16], revealing that immune enriched schwannomas classified as either inflammatory tumor microenvironment or reactive tumor microenvironment (Fig. 1a), consistent with the significant immune cell infiltration in these tumors. External validation methylation data was obtained from GSE79009[8] and preprocessed and normalized as above. The overlapping set of DNA methylation probes between the external validation cohort

(Illumina Methylation 450K array) and the 2000 differentially methylated probes from the discovery cohort (Illumina MethylationEPIC 850K array) was used for complete linkage hierarchical clustering of external validation cohort schwannomas with 1-Pearson correlation. Differential methylation analysis of clinical variables was performed by separating schwannoma methylation profiles based on binary variables (sex, prior surgery, prior radiotherapy) or bisecting the cohort at the median of continuous variables (size, growth rate, age) and performing differential methylation analysis as described above. DNA methylation probes at neural crest enhancers were identified by overlapping the coordinates of each methylation site with those of neural crest enhancers active in development[70].

To classify prospective vestibular schwannomas analyzed using single-cell RNA sequencing and CyTOF into molecular groups, a methylation-based classifier using differentially methylated probes from our discovery cohort of 66 tumors was used to construct a support vector machine (SVM) using *caret* version 6.0 in R version 3.6.0. A linear kernel SVM was constructed using training data comprising 75% of randomly selected samples from the discovery cohort with 10-fold cross validation. The top 2000 differentially methylated probes were used as variables. The model was applied to a test set comprising 25% of randomly selected samples from the discovery cohort, and receiver operating characteristics were acquired for 1000x resampling of test data. The SVM distinguished neural crest from immune-enriched schwannomas with 100% accuracy when used to classify randomly selected test sets of samples from our initial cohort (95% CI 78.2-100%, $p = 8.04 \times 10^{-5}$).

Copy number variant (CNV) calling was performed from schwannoma DNA methylation profiles, and revealed that loss of chromosome 22q, which contains the tumor suppressor *NF2*, was the most common CNV and occurred in 32% of tumors (Supplementary Data 1). To evaluate whether we could identify single nuclei CNVs from prospective schwannomas that demonstrated loss of chromosome 22q based on DNA methylation profiling, we applied CONICS[71], but were unable to identify a distinct tumor population harboring 22q loss, likely due, in part, to technical factors related to single-nuclei sequencing data from frozen specimens and minimal chromosomal instability of schwannomas[8,9]. Methylation arrays were deconvolved to identify tumor cell purity and constituent cell types (Supplementary Data 1) based on published approaches enabling comparison to a reference atlas of pure methylation profiles[72] or with a random forest classifier trained on the ABSOLUTE method developed by the TCGA for whole exome sequencing[73]. Using DNA methylation-based deconvolution, we compared immune-enriched tumors with a prior history of SRS to immune-enriched tumors without SRS, and identified no significant differences in the degree of immune (0.037 versus 0.025, p = 0.37), stroma (0.030 versus 0.010, p = 0.14) or microenvironment scores (0.067 vs. 0.036, p = 0.13).

## Fluorescence microscopy

Fluorescence microscopy was performed on a Zeiss LSM 800 confocal laser scanning microscope with Airyscan. Images were processed and quantified from at least 2 regions per condition using ImageJ[74]. Cilia prevalence was quantified as the ratio of cilia to nuclei, and ciliary fluorescence intensity was quantified from regions of interest normalization to background fluorescence.

## Histology and light microscopy

Formalin-fixed and paraffin-embedded whole slide sections were reviewed for histologic features including presence of capsule, cystic degeneration, Verocay bodies, biphasic morphology (Antoni A and Antoni B zones), degenerative atypia, hypercellular zones (bone-to-back or overlapping nuclei), hyalinized/thick-walled intratumor vessels, necrosis, hemosiderin deposition, calcification, hyalinization, and mitoses by a board-certified neuropathologist who was blinded to all

clinical and molecular data (MP). Mitotic activity was noted in regions of maximal activity as the number of mitotic figures per 10 consecutive high-power fields measuring 0.24 mm[75]. Inflammatory infiltrate including lymphocytes and macrophages were scored on H&E-stained sections as no immune cells, scattered immune cells, or abundant immune cells. All vestibular schwannomas with sufficient tissue were included in tissue microarrays (TMAs) containing 4 µm thick paraffin sections from 2 mm cores in duplicate for each case. In cases with biphasic histology, the cores predominantly targeted the cellular Antoni A zones and avoided areas with near-complete macrophage infiltrate or necrosis. All histologic features and stains were scored for each TMA core separately and were averaged for each case.

## Immunofluorescence and immunohistochemistry

Immunofluorescence for schwannoma SOX10 and cilia expression was performed using whole slide sections and TMAs. Sections were deparaffinized in xylene, rehydrated through graded ethanol dilutions and subjected to antigen retrieval using CC1 TRIS buffer (Ventana Medical Systems, #950-124); labeled with primary antibodies including SOX10 to mark schwannoma cells (API 3099, Biocare; labeling validated in schwannoma and melanoma), Pericentrin (PA5-54109, Thermo Fisher Scientific; labeling validated by Pericentrin knockdown) and γTubulin (T5192, Sigma; labeling validated in human and chicken cells) to mark centrosomes, and Acetylated Tubulin to mark cilia (T6793, Sigma; labeling validated in vertebrate and invertebrate organisms); labeled with Alexa Fluor secondary antibodies and DAPI to mark DNA (62248, Thermo Fisher Scientific); and mounted in ProLong Diamond Antifade Mountant (Thermo Fisher Scientific, #P36970).

Immunofluorescence for Human Schwann Cells (HSC) and Mouse Schwann Cells (MSC) cilia was performed on glass coverslips. Cells were fixed in 4% paraformaldehyde, blocked in 2.5% FBS, 200 mM glycine and 0.1% Triton X-100 in PBS for 30 min at room temperature (Thermo Fisher Scientific), and labeled with Smoothened (ab72130, Abcam; labeling validated by competition with immunizing peptide) and Centriolin (sc-365521, Santa Cruz Biotechnology; labeling validated using THP-1, SK-BR-3, and U-937 cell lysates) primary antibodies at 4 °C overnight. Cells were labeled with Alexa Fluor secondary antibodies, Hoescht to mark DNA (H3570, Life Technologies), and Acetylated Tubulin Alexa Fluor 647 Conjugate to mark cilia (sc-23950, Santa Cruz Biotechnology; labeling validated using A2058, 3T3-L1, and Jurkat cell lysates), for 1 hour at room temperature. Cells were mounted in Pro-Long Diamond Antifade Mountant (Thermo Fisher Scientific, #P36970).

Immunohistochemical stains for T cells and macrophages were performed using TMAs using standard techniques. In brief, slides were subjected to antigen retrieval and labeled with CD3 (A0452, Agilent Technologies, Santa Clara, CA; labeling validated by over-expression) or CD68 (M0814, Agilent Technologies; labeling validated using human B-cell lymphoma) primary antibodies. Detection was performed using the UltraView Universal DAB Detection Kit on the Ventana Benchmark Platform (Ventana Medical Systems, #760-500). CD3 was scored as the number of positive cells in each core as negative (≤10), low (11–50), moderate (51–100), and abundant (>100). CD68 was scored as negative (no positive cells recognizable at high power), weak staining in <10% of cells, moderate (staining in 10–20% of cells), or abundant (staining in >20% of cells, easily visible at low power). Immunohistochemistry for BCL1 (RM9104R7, Thermo Fisher Scientific labeling validated using MAD109 cell lysate) was performed as described for T cells and macrophages but on whole slide sections and with scoring as was done for CD68.

## Cell culture and treatments

Human Schwann cells (HSC, ScienCell Research Laboratories #1700) and Mouse Schwann Cells (MSC, ScienCell Research Laboratories #M1700-57) were cultured in complete Schwann Cell Medium on Poly-L-Lysine coated substrates (ScienCell Research Laboratories, #1701).

HEI-193 schwannoma cells were a gift from Marco Giovannini and cultured in Dulbecco's Modified Eagle Medium (Gibco, #11960069) supplemented with 10% fetal bovine serum (FBS) (Life Technologies, #16141), glutamine (Thermo Fisher Scientific, #10378016)[38]. HEK-293T cells were a gift from Luke Gilbert and cultured in Dulbecco's Modified Eagle Medium (Gibco, #11960069) supplemented with 10% fetal bovine serum (FBS) (Life Technologies, #16141). Cell cultures were authenticated by STR analysis at the UC Berkeley DNA Sequencing Facility, as well as routinely tested for mycoplasma using the MycoAlert Detection Kit (Lonza, #75866-212). Subconfluent HSC and HEI-193 schwannoma cells were irradiated with an X-Rad 320 (Precision X-Ray) irradiator using a 320 KV output at a rate of 3 Gy/min, with rotating a platform supporting cell culture plates. Cells were quantified by manual hemocytometer during routine cell culture passaging. Crystal violet staining of HEI-193 cells was quantified with background subtraction using ImageJ[74]. For ciliation and Hedgehog signaling assays, cultures were transitioned to OptiMEM (Thermo Fisher Scientific, #31985062) and treated with recombinant Sonic Hedgehog 1 µg/ml (1845, R&D Systems, Minneapolis, MN) or vehicle control for 24 h. Subconfluent HSC cultures in 96 well plates were treated with vismodegib (Genentech) for 72 h, and cell proliferation was assayed on a GloMax Discovery plate reader (Promega, #PAGM3000) using the CellTiter 96 Non-Radioactive Cell Proliferation kit (Promega, #G4100). For proteomic mass spectrometry, cells were grown in serum-free N5 media consisting of Neurobasal A (Life Technologies, #10888022) supplemented with N2 (Gemini Bio-Products, #400-163) and B27 supplements without vitamin A (Gibco, #12587010), L-glutamine 2 mM, antibiotic-antimycotic (Thermo Fisher Scientific, #15240112), bFGF 20 ng/mL (VWR, #119–126), and human EGF 20 ng/mL (Pepro-Tech, #AF-100-15). Cells were irradiated, and cell-free conditioned media was isolated by centrifugation for 10 min at 4000$g$ and filtering through a 0.45 µm syringe for mass spectrometry. TUNEL assays for apoptosis were performed using the APO-BrdU TUNEL Assay Kit with Alexa Fluor 488 Anti-BrdU (Thermo Fisher Scientific, #A23210).

## CRISPR interference

HEI-193 schwannoma cells stably expressing the CRISPRi components dCas9-KRAB were generated as previously described[14,40]. HEI-193 schwannoma cells were transduced with lentivirus harboring SFFV-dCas9-BFP-KRAB, and the top ~25% of cells expressing BFP were FACS sorted and expanded. For single gene targeted knockdowns, sgRNA protospacer sequences (Supplementary Data 11) were cloned into a lentiviral expression vector (U6-sgRNA EF1Alpha-puro-T2A-BFP) by annealing and ligation. HEI-193 schwannoma CRISPRi cells were then transduced with sgRNA lentivirus and selected with puromycin 1 µg/mL for at least 4 days before experimentation.

## Lymphocyte isolation and migration

Peripheral blood lymphocytes were isolated from the human blood of healthy volunteers[76]. A Polymorph density gradient (Accurate Chemical & Scientific Corporation, #AN221725) was used to isolate peripheral blood mononuclear cells that were subsequently selected and differentiated into T cell lymphocytes in Roswell Park Memorial Institute 1640 media (Life Technologies, #11875093) supplemented with 10% FBS, 1% penicillin/streptomycin, phytohemagglutinin 1 µg/ml (10576015, Thermo Fisher Scientific) and recombinant human IL-2 20 ng/ml (202-IL, R&D Systems). The QCM Leukocyte Migration Assay (MilliporeSigma, # ECM557) and a GloMax Discovery plate reader (Promega, #PAGM3000) were used to quantify transwell T cell lymphocyte migration from the apical chamber over 4 h at 37 °C. Unconditioned HEI-193 media served as a negative control, unconditioned HEI-193 media supplemented with recombinant human CCL21 600 ng/ml (366-6C, R&D Systems) served as a positive control, HEI-193 media 5 days after 12.5 Gy in 1 fraction, or HEI-193 media without radiation were placed in the basolateral chamber. All

conditions were free from serum or other supplements unless specifically indicated.

## Mass cytometry by time-of-flight cell preparation and analysis

All mass cytometry by time-of-flight (CyTOF) antibodies and concentrations used for analysis can be found in Supplementary Data 5. Primary conjugates of antibodies were prepared using the MaxPAR antibody conjugation kit (Fluidigm, #201153A). Following labeling, antibodies were diluted in Candor PBS Antibody Stabilization solution (Candor Bioscience, #131050) supplemented with 0.02% NaN3 to between 0.1 and 0.3 mg/mL final concentration and stored long-term at 4 °C. Each antibody clone and lot were titrated to optimal staining concentrations using primary human samples.

All tissue preparations were performed simultaneously from each sample, as previously described[29]. Tumors were finely minced and digested in L-15 medium with 800 units/ml collagenase IV (Worthington, #LS004186) and 0.1 mg/ml DNase I (Sigma). After digestion, re-suspended cells were quenched with PBS/EDTA at 4 °C. To determine viability, all tissues were washed with PBS/EDTA and re-suspended 1:1 with PBS/EDTA and 100 mM Cisplatin (Enzo Life Sciences, #ALX-400-040-M050) for 60 s before quenching 1:1 with PBS/EDTA/BSA[29]. Cells were centrifuged at $500g$ for 5 min at 4 °C and re-suspended in PBS/EDTA/BSA at a density between 1 and $10 \times 10^6$ cells/ml. Suspensions were fixed for 10 min at room temperature using 1.6% paraformaldehyde and frozen at −80 °C.

Mass-tag cell barcoding was performed as previously described[77]. Briefly, $10^6$ cells from each sample were barcoded with distinct combinations of stable Pd isotopes in 0.02% saponin in PBS. Samples were then barcoded together. Cells were washed once with cell staining media (PBS with 0.5% BSA and 0.02% NaN₃), once with 1× PBS, and pooled into a single FACS tube (BD Biosciences). After data collection, each condition was deconvoluted using a single-cell debarcoding algorithm[77].

Cells were resuspended in cell staining media comprised of PBS with 0.5% BSA and 0.02% NaN₃, and antibodies against CD16/32 were added at 20 mg/mL for 5 min at room temperature on a shaker to block Fc receptors. Surface marker antibodies were then added, yielding 500 μL final reaction volumes, and stained for 30 min at room temperature on a shaker. Following staining, cells were washed 2 times with cell staining media, permeabilized with permeabilization buffer (eBioscience, #00-8333-56), and stained with intracellular antibodies in 500 μL for 60 min at 4 °C on a shaker. Cells were washed twice in cell staining media and stained with 1 mL of 1:4000 191/193Ir DNA intercalator (Fluidigm, #201192A) and diluted in PBS with 1.6% paraformaldehyde overnight. Cells were washed once with cell staining media and 2 times with double-deionized water. Care was taken to assure buffers preceding analysis were not contaminated with metals in the mass range above 100 Da. Mass cytometry samples were diluted in Cell Acquisition Solution (Fluidigm, #201237) containing bead standards (see below) to approximately $10^6$ cells per mL and then analyzed on a CyTOF-2 mass cytometer (Fluidigm) equilibrated with Cell Acquisition Solution.

Data normalization was performed as previously described[77]. All mass cytometry files were normalized together using the mass cytometry data normalization algorithm[77], which uses the intensity values of a sliding window of bead standards to correct for instrument fluctuations over time and between samples. After normalization and debarcoding of files, singlets were gated by event length and DNA. Live cells were identified by cisplatin-negative cells. All positive populations, negative populations, and antibody staining concentrations were determined by titration on positive and negative control cell populations.

Scaffold maps were generated as previously described[29] using the open-source Statistical Scaffold R package available at github.com/SpitzerLab/statisticalScaffold. Landmark reference nodes were gated from peripheral blood mononuclear cells, while unsupervised clusters were generated via Clustering Large Applications (CLARA) clustering from samples pooled together, with each sample contributing an equal number of cells. Uniform manifold approximation and projection (UMAP) was performed on ArcSinh (cofactor = 5) transformed protein expression values on equal numbers of cells from each sample by randomly subsampling cells with parameters min.dist = 1.0. Clusters were identified by CLARA clustering using cells from all samples concatenated together.

## Proteomic mass spectrometry

HSC and HEI-193 media samples were prepared for mass spectrometry as described above, and 50 μL of media was mixed with 50 μL of lysis buffer containing 75 mM ammonium bicarbonate, 1% sodium deoxycholate, 5 mM TCEP and 40 mM chloroacetamide and incubated at 37 °C for 30 min to reduce and alkylate proteins. Proteins were digested with endoproteinase LysC (Wako Chemicals) and trypsin (Promega) overnight at 37 °C, followed by sodium deoxycholate and 2% TFA precipitation and centrifugation for 15 minutes at maximum speed. Peptides were desalted using MicroSpin Columns (The Nest Group) and resuspended in 4% formic acid and 3% acetonitrile. Approximately 1 μg of digested peptides per sample was loaded onto a 75 μm ID column packed with 25 cm of Reprosil C18 1.9 μm, 120 Å particles (Dr. Maisch GmbH HPLC), and eluted into an Orbitrap Fusion Lumos Tribrid mass spectrometer (Thermo Fisher Scientific) over the course of a 120-min acquisition by gradient elution delivered by an Easy1200 nLC system (Thermo Fisher Scientific). The composition of mobile phases A and B were 0.1% formic acid in water and 0.1% formic acid in 80% acetonitrile, respectively. All MS spectra were collected with orbitrap detection, and the most abundant ions were fragmented by higher energy collision dissociation and detected in the ion trap, with a 1-s cycle time between MS1 spectra. All data were searched against the UniProt human proteome database. Peptide and protein identification searches were performed using the MaxQuant data analysis algorithm, and all peptide and protein identifications were filtered to a 1% false discovery rate[78,79]. Label-free quantification and statistical testing were performed using the MSstats statistical R-package[80]. The mass spectrometry data files (raw and search results) have been deposited to the ProteomeXchange Consortium via the PRIDE partner repository with dataset identifier PXD014798[81]. Parallel reaction monitoring (PRM) measurements were acquired by LC−MS/MS on a Q-Exactive Plus (Thermo Fisher Scientific, IQLAAEGAAPFALGMBDK) mass spectrometer equipped with an EASY-nLC 1200 system (Thermo Fisher Scientific, #LC140) using the same column configuration and HPLC settings as described above. All PRM data was analyzed by Skyline 4.2.0.19072[82], quantified via MSstats, and have been deposited to Panorama Public[83] with dataset identifier PXD014883.

## Targeted metabolite liquid chromatography−mass spectrometry

HEI-193 cells were cultured and treated as described above, and 100 μL of cell-free supernatant (CFS) or cell pellets containing 10 million cells each were separately combined with cold $H_2O$:MeOH at a ratio of 1:1 containing internal standards of 2-amino-3-bromo-5-methylbenzoic acid and a labeled amino acid mixture (13C,15N) spiked-in at 1 μg/mL and 10 nM/mL, respectively. CFS and cell pellets were briefly vortexed or rotated overnight, respectively, and immediately chilled at 4 °C. All samples were centrifuged at 4 °C for 30 min at maximum speed, and the supernatant was aliquoted for quantification.

Targeted liquid chromatography-high resolution mass spectrometry analysis was performed on a Sciex Exion UPLC equipped with a C18 Polar column (Phenomenex, 150 × 2.1 mm, 2.6 μm) and coupled to a Sciex QTRAP 7500 operated in negative mode with MRM acquisition of MS/MS data. A gradient of 0−90% acetonitrile over 16 min was used for compound separation. The XICs of all samples and QCs were visually inspected. All QCs displayed appropriate signals for analysis in

addition to a significant peak capacity and retention time reproducibility. Raw data files were uploaded to SciExOS for processing, where filtering, feature detection, integration, and alignment of the chromatograms in each sample were performed. As a result, data matrices were generated with intensity values representing the area under the curve of significant chromatographic peaks above the lower limit of detection.

## Primary schwannoma cell culture, treatment, and metabolomic mass spectrometry

Primary human vestibular schwannomas were cultured as previously described[84]. Tumors were collected from the operating room and placed on ice for transfer to the lab. Under sterile conditions in a cell culture hood, the tissue was minced into 1 mm³ piece and then digested for 45 min at 37 °C in 0.25% trypsin/0.1% collagenase solution. Cells were resuspended in Dulbecco's Modified Eagle Medium (Gibco, #11960069) supplemented with insulin (10 μg/mL, Sigma), 10% fetal bovine serum, and N2 supplement (Gemini Bio-Products, #400-163). The cell suspension was plated onto culture dishes pre-treated with poly-ʟ-ornithine and laminin. Cultures were placed in a humidified incubator at 37 °C with 5.0% $CO_2$ and grown for 7–14 days without passage. Media was exchanged every 2–3 days. Cultures were treated with 0, 3, 10, or 20 Gy single fraction gamma-irradiation from a radioactive Cesium source. Six or 72 hours after radiotherapy, cultures were washed with ice-cold PBS and water and flash-frozen in liquid nitrogen. Derivatized metabolite extracts from these primary schwannoma cell cultures were analyzed by gas chromatography–mass spectrometry on a ThermoISQ Quadrupole. Raw data were analyzed with TraceFinder 4.1 (Thermo). Cell culture data were normalized to the total ion signal to control for extraction, derivatization, and/or loading effects.

## Quantitative polymerase chain reaction

RNA was isolated using the RNEasy Mini Kit (Qiagen, #74106) and a QiaCube (Qiagen, #9001292), and cDNA was synthesized using the iScript cDNA Synthesis Kit (Bio-Rad, #1708891) and a ProFlex thermocycler (Thermo Fisher Scientific, #4484073). Target genes were amplified using PowerUp SYBR Green Master Mix (Thermo Fisher Scientific, #A25741) and a QuantStudio 6 thermocycler (Thermo Fisher Scientific, #4485691). Gene expression was calculated using the ΔΔCt method for candidate genes, with normalization to *GAPDH* (Supplementary Data 11).

## Bulk RNA sequencing and analysis

RNA was isolated from frozen tumors or cell cultures using the All-Prep Universal Kit, and clean-up was performed using the RNEasy Kit as needed (Qiagen). RNA sequencing libraries were generated using the Illumina TruSeq Stranded mRNA Library Prep Kit and sequenced on an Illumina HiSeq-4000 using the paired-end 100 protocol for schwannomas and single-end 50 protocol for HEI-193 schwannoma cells. Reads were aligned to GRCh38 using the splice-aware aligner HISAT2 version 2.0.3 against an index containing SNP and transcript information (genome_snp_tran)[85]. Ensembl build 75 genes were quantified with featureCounts using uniquely mapped reads[86]. Differential expression analysis was performed using DESeq2 using the Wald test with an adjusted *p*-value threshold of 0.05 corrected for multiple hypotheses using the Benjamini-Hochberg method[87]. Complete linkage hierarchical clustering was performed using 1-Pearson correlation as the distance matrix with differentially expressed genes.

## Single-cell RNA sequencing and analysis

HEI-193 schwannoma cells were dissociated in Trypsin–EDTA 0.25% (Thermo Fisher Scientific, #25200114), passed through a 40 μm strainer and washed/resuspended in phosphate-buffered saline. Fresh schwannomas were minced with sterile Bard-Parker #10 surgical scalpels (Aspen Surgical, #4–410) and incubated in 0.4% Collagenase Type

2 (Worthington, # LS004174) in pre-oxygenated Dulbecco's Modified Eagle Medium (Thermo Fisher Scientific, #11960069) for 75 min at 37 °C while rotating at 800 rotations per minute on a thermomixer. The suspension was sequentially filtered through 70 μm (Corning, #352350) and 40 μm strainers (Corning, 352340), centrifuged at 300*g* for 5 min, and resuspended in cold phosphate-buffered saline. Single-cell suspensions were loaded onto a 10× Chromium controller using the Chromium Single Cell 3' Library & Gel Bead Kit v3 (10× Genomics). Libraries were sequenced on an Illumina NovaSeq (10× specific protocol) with >50,000 reads per cell.

Library demultiplexing, read alignment to human genome GRCh38, and unique molecular identifier (UMI) quantification was performed in Cell Ranger version 1.3.1 (10× Genomics). Schwannoma cells with greater than 200 unique genes were detected, and fewer than 15% of reads attributed to mitochondrial transcripts were retained. Data were normalized, and variance stabilized by SCTransform in Seurat version 3.0 using UMI count and percent of reads aligned to mitochondrial transcripts as covariates[88]. UMAP was performed on significant ($p < 0.05$) principal components (determined by JackStraw analysis) and batch-corrected using Harmony with parameters min.dist = 0.3. Louvain clustering was performed with resolution = 0.3, and cluster markers were identified based on expression in at least 25% of cells and differential expression by more than 25% compared to all other clusters. For HEI-193 schwannoma cells, analysis was performed as above, except that cells with greater than 2000 UMIs and fewer than 20% of reads attributed to mitochondrial transcripts were retained, and UMAP was performed with min.dist = 0.2 without harmonization. Gene ontology analysis was performed using Enrichr without the inclusion of ribosomal subunits or mitochondrial genes[69]. Cell cluster identification was further analyzed using the AddModuleScore function within Seurat 3.0 to score expression signatures based on gene sets from the Molecular Signatures Database (MSigDB)[89], intersected with differentially expressed cluster markers described above, and then normalized across all populations in UMAP space[88].

## Single-nuclei RNA sequencing and analysis

Flash-frozen archived schwannoma specimens were minced with sterile Bard-Parker #10 surgical scalpels (Aspen Surgical, #4–410) and mechanically dissociated with a Pestle Tissue Grinder (size A, Thomas Scientific, #3431E45) in ice-cold lysis buffer consisting of 0.32 M sucrose, 5 mM CaCl2, 3 mM MgAc2, 0.1 mM EDTA, 10 mM Tris-HCl, 1 mM DTT and 0.1% Triton X-100 in DEPC-treated water[90]. A sucrose solution consisting of 1.8 M sucrose, 3 mM MgAc2, 1 mM DTT, and 10 mM Tris-HCl in DEPC-treated water was added to the bottom of the lysis solution in ultracentrifuge tubes (Beckman Coulter) to form a gradient, which was ultracentrifuged at 107,000*g* for 2.5 h at 4 °C. Nuclei pellets were resuspended in phosphate-buffered saline and sequentially filtered twice in 30 μm strainers (Miltenyi Biotec, #130-098-458). Isolated nuclei were assessed with DAPI staining and loaded onto a 10× Chromium controller using the Chromium Single Cell 3' Library & Gel Bead Kit v2 (10× Genomics). Library sequencing and preprocessing for single nuclei were performed as described above for single-cell libraries, except that a pre-mRNA reference library (GRCh38), including intronic segments, was used for read alignment and quantification. Nuclei libraries with greater than 400 unique genes were detected, and fewer than 5% of reads attributed to mitochondrial transcripts were retained.

To integrate single-nuclei and single-cell RNA sequencing data, all libraries passing respective QC filters described above were combined in silico and normalized with variance stabilization using SCTransform in Seurat version 3.0, with UMI count, percent of reads aligned to mitochondrial transcripts, and technique (single-cell versus single-nuclei) as covariates[88]. Principal component analysis was performed, and single nuclei and single-cell data were harmonized using Harmony in Seurat version 3.0 using technique and day-of tumor isolation as covariates. Clustering and marker identification was performed as

described above for single cell-only analysis, with parameters min.dist = 0.3 and resolution = 0.3.

## CRISPR interference genome-wide screening

CRISPRi screens were performed as described previously[14,15]. HEI-193 cells stably expressing CRISPRi components (dCas9-KRAB) were transduced with lentivirus supernatant containing the third-generation dual sgRNA CRISPRi library, which targets 20,528 genes and 1025 sgNTC[15]. Screens were performed in triplicate cultures with coverage of at least 500× cells per construct. sgRNA-expressing cells were selected using puromycin (1 μm/mL) for 48 h and transferred to puromycin-free normal growth media for 48 h to allow recovery. Initial (T0) cell populations were then frozen in 10% DMSO and processed for genomic DNA alongside endpoint (T12) cell populations, which corresponded to 7.82 population doublings. Triplicate screens were also performed with radiotherapy (1.8 Gy × 5 fractions) delivered daily starting on T0, with the endpoints (T12) corresponding to 3.46 population doublings. Genomic DNA was harvested using the NucleoSpin Blood L Kit (Machery-Nagel, #740954.20) for each cell population, and sgRNA cassettes were amplified using 22 cycles of PCR using NEBNext Ultra II Q5 PCR MasterMix (New England Biolabs, #M0544L). Sequencing was performed on a NovaSeq 6000 (Illumina) using custom sequencing primers[15].

sgRNA read counts were aligned using custom Python scripts derived from the ScreenProcessing package[15], without allowing mismatches. sgRNA counts with discordant target genes from the same vector, representative of vector recombination, or fewer than 100 reads detected in the T0 populations, were discarded from downstream analysis. Growth phenotype (gamma) was defined as log2(sgRNA count T12/sgRNA count T0) minus median sgNTC log2(sgRNA count T12/sgRNA count T0) as previously described[14]. Radiation phenotype (rho) was defined as log2(sgRNA count T12 (1.8 Gy × 5)/sgRNA count T12 (0 Gy)). Statistical significance was quantified using a two-sided Student's $t$-test comparing replicate distributions of library-normalized counts for each sgRNA between conditions (rho) or time points (gamma). A discriminant threshold of 5, derived from the product of normalized gene-phenotype and $-\log10(p$-value), corresponding to an empiric false discovery rate of ~1%, was selected for hit definitions[14]. To ascertain the fidelity of our screens, we overlapped all negative growth hits from our screen (without radiotherapy treatment) with common essential genes from DepMap[91] and found that 591 out of 918 total negative growth hits (64.4%) were also DepMap essential genes. The reproducibility of our screens was assessed by calculating the Pearson correlation coefficients of the screen hit phenotypes, which demonstrated a median R of 0.829 for no radiotherapy and 0.760 for radiotherapy conditions.

## Perturb-seq

The single-cell Perturb-seq library was composed of sgRNAs targeting genes from RNA sequencing and DNA methylation profiling of human vestibular schwannomas (Supplementary Data 11). Using a single sgRNA vector strategy, protospacer sequences were selected from the optimized human CRISPRi v2.1 library[92]. Library cloning was performed as previously described[93]. Insert lyophilized oligonucleotides (Twist Biosciences) containing the sgRNA sequences were resuspended at 100 nM in H$_2$O and amplified by PCR using HF Phusion polymerase (New England Biolabs, #M0531S). After verifying amplification on a 10% acrylamide gel, the PCR product was purified using MinElute Cleanup Kit (Qiagen, #28206). The pBA904 Perturb-seq sgRNA vector backbone (Addgene, #122238) was digested using FastDigest BstX1 (ThermoFisher Scientific, #FD1024) and Blp1 (ThermoFisher Scientific, #FD0094) and excised from a 0.8% agarose gel for purification using the NucleoSpin Gel Cleanup Kit (Macherey-Nagel, #740609.50). The insert pool and digested backbone were then ligated overnight at 16 °C using T4 DNA Ligase (New England Biolabs, #M0202T) with a vector-to-insert ratio of

1:1. The ligation reaction was purified by ethanol precipitation and transformed into Stellar Competent Cells (Takara, #636766) to assess library diversity by Sanger sequencing of 10 clones. For large-scale transformation, the library was transformed into MegaX Electrocompetent cells (ThermoFisher Scientific, #C640003) using electroporation on 15-cm LB carbenicillin plates. The library stock was prepared by scraping plates, followed by purification using the Midi Prep kit (Macherey-Nagel, #740410.50). Library sequencing was performed on an Illumina MiSeq run to ensure sgRNA uniformity. Using lentivirus, cells were transduced with the library at 1000× coverage at an MOI of 0.1, corresponding to approximately 95% of cells with a single integration. Seventy-two hours following transduction, sgRNA+ cells were FACS sorted (BFP+) and were recovered in DMEM 10% FBS. Sorted cultures were treated with either 0 Gy, 1.8 Gy × 5 daily fractions, or 12.5 Gy × 1 fraction of radiotherapy using the X-Rad 320 irradiator (Precision X-Ray). Twelve hours following the final fraction of radiotherapy, cells were trypsinized and harvested in single-cell suspension on the 10× Chromium Controller (10× Genomics, #1000204).

Single-cell Perturb-seq libraries were processed using the Chromium Next GEM Single Cell 3' GEM, Library & Gel Bead Kit v3.1 with Feature Barcoding (10× Genomics, #1000269), allowing direct capture of modified sgRNAs[93] and sequenced on an Illumina NovaSeq-6000. Cells from the 0 Gy condition were run across 2 GEM groups, while each of the radiation conditions was run on its own GEM group. Library demultiplexing, gene expression read alignment to human genome GRCh38, UMI quantification, and sgRNA assignment and quantification were performed in Cell Ranger version 6.1.2 with sgRNA barcoding (10× Genomics).

Data analysis was performed using Seurat 4.3.0 in R version 4.2.2. Cells with greater than 200 detected features and UMIs mapping to only one sgRNA were retained. Target gene knockdown was quantified by library normalizing the transcriptome of each cell and obtaining the mean expression of each gene across all cells belonging to a given sgRNA within an individual GEM group (pseudobulk). RNA remaining for each gene target was calculated by dividing the pseudobulk expression into on-target cells with those of sgNTC cells, with a pseudo count of 0.01 added to each component. Only sgRNAs exerting greater than 75% knockdown in at least one of the two 0 Gy conditions, and also represented in at least 10 cells in each condition, were retained for analysis. To score expression changes of gene modules, marker genes were derived from Louvain clustering of human schwannoma cell types (Fig. 1b) and HEI-193 schwannoma cell states (Fig. 2c), as described above. The top 10 most specific cluster markers were used to interrogate each cell type or cell state, and the mean pseudobulk expression of these markers was obtained for each on-target sgRNA as well as for sgNTC cells, in each radiotherapy condition. sgRNA phenotypes were then normalized to pseudobulk cells containing sgNTC in the 0 Gy condition. These fold changes were log$_2$ transformed and grouped within radiotherapy conditions using hierarchical clustering with Pearson correlation and complete linkage.

Differential expression analysis in Perturb-seq data was performed using the FindMarkers function in Seurat with MAST as the test type. Genes with $p < 0.05$ and |log$_2$(fold change)| > 1) were considered differentially expressed and used for gene ontology analyses in EnrichR with separate queries for all enriched or suppressed genes. To identify perturbations that preferentially generated phenotypes in radiotherapy conditions, the number of significantly differentially expressed genes was compared to identify perturbations with more than 40 excess differentially expressed genes in the radiotherapy (1.8 Gy × 5) conditions compared to control (0 Gy) conditions.

## Single-nuclei ATAC, RNA, and CRISPRi perturbation sequencing (snARC-seq)

Genetic perturbations in snARC-seq rely on the CROP-seq vector (Addgene, pBA950)[94], which allows for the capture of sgRNA identity

from nuclear RNA transcripts. This design permits single sgRNA perturbations, and therefore we designed 2 independent sgRNA vectors per target gene, corresponding to each of the dual sgRNAs that targeted hit genes from the genome-wide CRISPRi V3 library[15]. Oligonucleotides containing protospacer sequences (Supplementary Data 11) were ordered as a pool from Twist Bioscience and PCR amplified as described above. The oligonucleotide pool was ligated into the CROP-seq backbone using BstXI and BlpI digestion and T4 ligation. Library representation, including sgNTC overrepresentation, was performed on a MiSeq run to ensure sgRNA uniformity.

The cloned library was packaged into lentivirus using HEK-293T cells. HEI-193 cultures were transduced to an MOI of 0.1, and FACS sorted for sgRNA+ cells after 48 h. Radiotherapy was then delivered to either 0 Gy or 1.8 Gy × 5 daily fractions using the X-Rad 320 irradiator (Precision X-Ray). Following completion of radiotherapy, cells were harvested with Trypsin and prepared following the Nuclei Isolation for Single Cell Multiome ATAC + Gene Expression Sequencing 10× Protocol (10× Genomics, CG000365 Rev C). Briefly, $1-2 \times 10^5$ cells were incubated in chilled Lysis Buffer (Tris-HCl (pH 7.4) 10 mM, NaCl 10 mM, MgCl$_2$ 3 mM, Tween-20 0.1%, Nonidet P40 Substitute 0.1%, Digitonin 0.01%, BSA 1%, DTT 1 mM, RNase inhibitor 1 U/μL) on ice for 4 min, washed with Wash Buffer (Tris-HCl (pH 7.4) 10 mM, NaCl 10 mM, MgCl$_2$ 3 mM, Tween-20 0.1%, BSA 1%, DTT 1 mM, RNase inhibitor 1 U/uL) 3 times, and resuspended in diluted nuclei buffer. Nuclei were counted, and membrane integrity was evaluated by Trypan staining on the Countess II FL Automated Cell Counter (Thermo, #TF-CACC2FL). Nuclei suspensions were diluted to approximately 3000 nuclei/μL and processed according to the 10× Chromium Next GEM Single Cell Multiome ATAC + Gene Expression protocol (CG000338 Rev A).

To recover sgRNA identities from single-nuclei RNA fractions, CROP-seq guides were amplified into dual-indexed Illumina libraries from the cDNA product of the 10× multiome protocol as described above with a three-round hemi-nested PCR as previously described[95]. Totally, 15 ng of full-length cDNA product was amplified with primers binding to the sgRNA constant region and 10× Genomics Read 1 Adapter. Two subsequent PCR steps were performed to introduce i5 and i7 indices and Illumina P5 and P7 adapters. After the first round of PCR, each product was size-selected using SPRI beads at 1.0×. Subsequent PCRs were conducted with 1 ng product. The final PCR product was size selected with 0.5×, and then 1.0× SPRI beads, and library quality was assessed by TapeStation high-sensitivity D1000 analysis (Thermo, #5067-5584). CROP-seq libraries were pooled with gene expression libraries and sequenced on an Illumina NovaSeq 6000 using the paired-end 100 bp protocol.

Library demultiplexing, read alignment to human genome GRCh38, and UMI quantification for the RNA and ATAC fractions were performed using Cell Ranger ARC version 2.0.1 (10× Genomics). Crop-seq sgRNAs were detected using kallisto bustools (v0.24.1)[96]. First, a sgRNA reference was built using kb ref with a k-mer size of 15 which was used as a reference for kb count. The sgRNA enrichment PCR sequences were then pseudo-aligned to this index using kallisto bustools, including the ARC multiome Gene Expression Whitelist version 1 (10× Genomics). For each cell barcode group, sgRNAs with less than 6 UMIs were filtered, and sgRNAs were assigned to cells using Geomux (v0.2.1)[54] (https://github.com/noamteyssier/geomux), which performs a hypergeometric test for each cell on its observed guide counts, then calculates a log$_2$-odds ratio between the two highest counts. Cells were assigned to their majority guide if their Benjamini–Hochberg corrected $P$ value was below 0.05, the log-odds ratio was above 1, and the total number of UMIs was greater than 6. The resulting sgRNA assignments were used to determine cell barcode groups with a single detected sgRNA.

Preprocessing of single-nuclei RNA and ATAC data was performed using Signac v1.8.0[97]. Cells containing the following quality measures were retained: ATAC UMI between 1000 and 100,000, RNA UMI between 50 and 25,000, nucleosome signal <4, and TSS enrichment >1.

ATAC peak calling was performed using the MACS2 wrapper in Signac. Peaks were processed using term frequency-inverse document frequency (TF-IDF) normalization and singular value decomposition. RNA counts were normalized, and variance was stabilized by SCTransform using default parameters. ATAC UMAP projection was calculated using the latent semantic indexing reduction dimensions 2–30. RNA UMAP projection was calculated using principal component analysis dimensions 1:20, which was empirically determined using ElbowPlot. Louvain clustering for either ATAC or RNA data was performed using a resolution parameter of 0.5.

Gene activity scores from the ATAC signal were generated by quantifying ATAC UMIs mapped to the promoter of each gene, defined as between the TSS and 2000 bp upstream of the TSS for each gene. To score gene activity changes of gene modules between radiotherapy and control conditions, gene activity scores were calculated for marker genes derived from Louvain clustering of human schwannoma cell types (Fig. 1b) and HEI-193 schwannoma cell states (Fig. 2c), as described above. Expression and gene activity were then pseudo-bulked according to the target gene and therefore included cells with either of the two targeting sgRNAs. The top ten most specific cluster markers were used for each cell type or cell state, and the mean pseudobulk gene activity of these markers was obtained for each on-target sgRNA as well as for sgNTC cells, in each radiotherapy condition. sgRNA phenotypes were then log$_2$ transformed, and the gene activity vector in radiotherapy (1.8 Gy × 5) conditions were subtracted by those in control conditions (0 Gy). These fold changes were clustered using hierarchical clustering with Pearson correlation and complete linkage.

Transcription factor motif deviations in the setting of genetic or therapeutic perturbations were quantified using the ChromVAR[59] wrapper in Signac with default parameters against the peaks assay. Mean motif deviations for each sgRNA identity in each condition were quantified and subsetted to only motifs whose cognate transcription factors were expressed in HEI-193 cells using RNA sequencing data, as described above. To estimate the differential motif deviations for a given sgRNA perturbation in radiotherapy conditions versus control conditions, deviations were log$_2$ transformed, and deviations in radiotherapy conditions were subtracted by those in control conditions. To further quantify chromatin accessibility at motif regions as a consequence of epigenetic regulator snARC-seq perturbations, differentially accessible regions were determined using the FindMarkers function in Signac using logistic regression. Differentially accessible regions were further subsetted by those whose nearest neighbor gene was associated with ChIP-seq peaks derived from the ENCODE project (KLF13 accession ENCFF453MMH, SETDB1 accession ENCFF773RNU). Average ATAC profiles were obtained for each sgRNA condition at these restricted sets of differentially accessible regions with a window ±1000 bp and then normalized to the average read depth at 100 bps at each end flank of the distributions.

## Statistics

All experiments were performed with independent biological replicates and repeated, and statistics were derived from biological replicates. Biological replicates are indicated in each figure panel or figure legend. No statistical methods were used to predetermine sample sizes, but sample sizes in this study are similar or larger to those reported in previous publications. Data distribution was assumed to be normal, but this was not formally tested. Investigators were blinded to conditions during clinical data collection and analysis of mechanistic or functional studies. Bioinformatic analyses were performed blind to clinical features, outcomes, or molecular characteristics. The clinical samples used in this study were retrospective and nonrandomized with no intervention, and all samples were interrogated equally. Thus, controlling for covariates among clinical samples is not relevant. Cells were randomized to experimental conditions. No clinical, molecular, or cellular data points were excluded from the analyses. Unless

specified otherwise, lines represent means, and error bars represent the standard error of the means. In general, statistical significance is shown by asterisks (*$p < 0.05$, **$p \leq 0.01$, ***$p \leq 0.0001$), but exact $p$-values are provided in the figure legends when possible. Lines show means ± standard error of the means. Boxplots show the 1st quartile, median, and 3rd quartile; whiskers represent a 1.5 inter-quartile range, and data outside this range are shown by points. Results were compared using Student's $t$-tests, Fisher's exact tests, and ANOVA, which are indicated in figure legends alongside approaches used to adjust for multiple comparisons. Multiple linear regression was performed in $R$ (v 3.6.0). Recursive portioning analysis was performed using the *rpart* package in $R$, with a minimum split size of 10 and a complexity parameter of 0.02. Variables with a $p$-value of 0.1 or less on univariate analysis were included in RPA and multivariate analysis. Logistic regression models were constructed and evaluated using receiver operating characteristic analysis. A nomogram was created using the *rms* package in R, which generates a graphical nomogram based on a general linear model.

## Reporting summary

Further information on research design is available in the Nature Portfolio Reporting Summary linked to this article.

## Data availability

DNA methylation array data are deposited on GEO under accession GSE222042. The Hg19 human reference genome was used to analyze DNA methylation arrays [https://bioconductor.org/packages/IlluminaHumanMethylationEPICanno.ilm10b2.hg19/]. All sequencing raw data, including bulk RNA-seq, scRNA-seq, snRNA-seq, Perturb-seq, and snARC-seq using the Illumina Hiseq 4000 and Novaseq 6000 sequencers, are deposited on SRA under accession PRJNA917076. Proteomic mass spectrometry data have been deposited on ProteomeXchange under accession PXD014798. Metabolomic mass spectrometry data have been deposited on MassIVE under accession MSV000091760. CyTOF raw data are deposited on [https://doi.org/10.17632/hjmvnf48gh.1][98]. Source data are provided in this paper.

## Code availability

Computational code for data analysis is deposited on https://github.com/liujohn/schwannoma [https://doi.org/10.24433/CO.0193242.v1][99].

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

## Acknowledgements

The authors thank Jeremy Reiter, Monika Sigg, Aparna Bhaduri, Aaron Tward, and Zora Arum for providing comments and reagents; Eric Chow, Derek Bogdanoff, Tomasz Nowakowski, Chang Kim, and Galina Schmunk for technical sequencing assistance and analysis; and Ken Probst and Noel Sirivansanti for Illustrations. This study was supported by the ASCO Conquer Cancer Sontag Foundation Young Investigator Award to SJL; NIH grants R21 HD106238 and R01 CA251221, and DOD grant NFRP NF200021 to D.R.R.; NIH grant U54 CA209891 to N.J.K.; a Children's Tumor Foundation Young Investigator Award to H.N.V.; and an American Otological Society Fellowship Grant to M.C.D. Sequencing was performed at the UCSF CAT, supported by UCSF PBBR, RRP IMIA, and NIH grant S10 OD028511.

## Author contributions
All authors made substantial contributions to the conception or design of the study; the acquisition, analysis, or interpretation of data; or drafting or revising the paper. All authors approved the paper. All authors agree to be personally accountable for individual contributions and to ensure that questions related to the accuracy or integrity of any part of the work are appropriately investigated, resolved, and the resolution documented in the literature. S.J.L., N.W.C., M.P., M.C.D., J.E.V.M., D.L.S., H.N.V., U.E.L., C.D.E., N.J.K., M.H., M.H.S., L.G., P.V.T., and D.R.R. designed the study and analyses. Experiments were performed by S.J.L., T.C.C., N.W.C., J.S., M.P., M.C.D., K.F., W.C.C., D.L.S., H.N.V., A.C., J.P., J.D.B., U.E.L., K.J.H.G., E.S., K.H.C., B.V.L., D.W., and D.R.R. Data analysis was performed by S.J.L., T.C.C., N.W.C., J.S., M.P., M.C.D., W.C.C., J.E.V.M., D.L.S., H.N.V., U.E.L., and D.R.R. The study was supervised by S.J.L., H.N.V., S.E.B., P.K.S., S.T.M., D.L., M.W.M., M.S.B., A.P., N.J.K., M.H., M.H.S., L.G., P.V.T., and D.R.R. The paper was prepared by S.J.L. and D.R.R. with input from all authors.

## Competing interests
The authors declare no competing interests.

## Additional information

[1]Department of Radiation Oncology, University of California San Francisco, San Francisco, CA 94143, USA. [2]Department of Neurological Surgery, University of California San Francisco, San Francisco, CA 94143, USA. [3]Department of Pathology, University of California San Francisco, San Francisco, CA 94143, USA. [4]Arc Institute, Palo Alto, CA 94304, USA. [5]Parker Institute for Cancer Immunotherapy, Chan Zuckerberg Biohub, and Departments of Otolaryngology, and Microbiology and Immunology, University of California San Francisco, San Francisco, CA 94115, USA. [6]Department of Urology, University of California San Francisco, San Francisco, CA 94143, USA. [7]Departments of Otolaryngology and Neurosurgery, University of Iowa, Iowa City, IA 52242, USA. [8]Department of Radiology and Biomedical Imaging, University of California San Francisco, San Francisco, CA, USA. [9]J. David Gladstone Institutes, California Institute for Quantitative Biosciences, Department of Cellular and Molecular Pharmacology, University of California San Francisco, San Francisco, CA 94158, USA. [10]Department of Dermatology, University of California San Francisco, San Francisco, CA 94115, USA. [11]Department of Neurological Surgery, Northwestern University, Chicago, IL 60611, USA. [12]Baptist Health Miami Neuroscience Institute, Miami, FL 33176, USA. [13]These authors contributed equally: Tim Casey-Clyde, Nam Woo Cho. ✉e-mail: david.raleigh@ucsf.edu

