## [Peer review file · Nature Communications]

REVIEWERS' COMMENTS

Reviewer #2 (Remarks to the Author):

The authors have addressed my previous concerns.

Reviewer #3 (Remarks to the Author):

The authors have made impressive revisions including the development of a very interesting technique Single-nuclei ATAC, RNA, CRISPRi perturbation sequencing (snARC-seq) that will be useful for other research groups. As compared to their initial submission, there is no more overstatement and all conclusions are well supported by the presented data. This is a highly interesting study that should be published now in the current form. I only have one minor comment:

Page 4, 2nd paragraph: as I previously mentioned in my initial review, Sox6 has not been shown to inhibit Schwann cell differentiation, but instead to inhibit oligodendrocyte differentiation. Now, the authors have changed the reference, as I requested, but they also need to change the text and remove the statement that Sox6 inhibits Schwann cell differentiation... Sox5 and Sox6 are known inhibitors of oligodendrocyte differentiation, but in Schwann cells, Sox5 is a mild inducer of differentiation and Sox6 is not expressed (Ittner et al, Sci. Rep., 2021).

Reviewer #4 (Remarks to the Author):

The authors have largely addressed all of the comments from the original submission. However, it is highly recommended that they not use the term “immunogenic schwannoma cells” based solely on transcriptomal data. In order to employ such a term, which implies function, they would need to perform experiments to demonstrate that they either function as antigen presenting cells, are targets of T cells, or the like.

Reviewer #5 (Remarks to the Author): Expert in brain tumours, ATAC-seq, epigenetics, and functional genomics; replaced the original Reviewer #1

This work investigates the effects of ionizing radiation on the epigenetic interconversion of schwannoma molecular groups in a thorough manner that addresses most (if not all) of the previous reviewers' concerns. Given the rarity of the disease, I must firstly commend the authors on their considerable effort to increase the power of their analyses and process additional samples prior to the resubmission. The manuscript appears to have undergone a major overhaul, including new techniques for data interrogation. The addition of CRISPRi-based experiments to identify the epigenetic regulators of Schwannoma cell reprogramming provides the mechanistic validation that was previously missing and strengthens their more conservative conclusions. I would have appreciated greater emphasis on the translational potential of their findings rather than their paradigm but still recognize its significance in the field of cancer research and recommend it for publication.

Reviewer #2

The authors have addressed my previous concerns.

Thank you.

Reviewer #3

The authors have made impressive revisions including the development of a very interesting technique Single-nuclei ATAC, RNA, CRISPRi perturbation sequencing (snARC-seq) that will be useful for other research groups. As compared to their initial submission, there is no more overstatement and all conclusions are well supported by the presented data. This is a highly interesting study that should be published now in the current form. I only have one minor comment: Page 4, 2nd paragraph: as I previously mentioned in my initial review, Sox6 has not been shown to inhibit Schwann cell differentiation, but instead to inhibit oligodendrocyte differentiation. Now, the authors have changed the reference, as I requested, but they also need to change the text and remove the statement that Sox6 inhibits Schwann cell differentiation... Sox5 and Sox6 are known inhibitors of oligodendrocyte differentiation, but in Schwann cells, Sox5 is a mild inducer of differentiation and Sox6 is not expressed (Ittner et al, Sci. Rep., 2021).

We have changed the text and removed the statement that SOX6 inhibits Schwann cell differentiation and instead now state that SOX6 is a “inhibitor of oligodendrocyte cell differentiation.”

Reviewer #4

The authors have largely addressed all of the comments from the original submission. However, it is highly recommended that they not use the term “immunogenic schwannoma cells” based solely on transcriptomal data. In order to employ such a term, which implies function, they would need to perform experiments to demonstrate that they either function as antigen presenting cells, are targets of T cells, or the like.

We have changed the term “immunogenic schwannoma cells” to “immune-like” schwannoma cells. We also note that Schwann cells have been previously described as capable of antigen presentation and T cell interaction (PMID: 28970572), as well as modulate local immune responses (PMID: 17823955).

Reviewer #5

This work investigates the effects of ionizing radiation on the epigenetic interconversion of schwannoma molecular groups in a thorough manner that addresses most (if not all) of the previous reviewers’ concerns. Given the rarity of the disease, I must firstly commend the authors on their considerable effort to increase the power of their analyses and process additional samples prior to the resubmission. The manuscript appears to have undergone a major overhaul, including new techniques for data interrogation. The addition of CRISPRi-based experiments to identify the epigenetic regulators of Schwannoma cell reprogramming provides the mechanistic validation that was previously missing and strengthens their more conservative conclusions. I would have appreciated greater emphasis on the translational potential of their findings rather than their paradigm but still recognize its significance in the field of cancer research and recommend it for publication.

We appreciate the reviewer’s appraisal of our revised manuscript.